# Oxytocin receptor DNA methylation is associated with exogenous oxytocin needs during parturition and postpartum hemorrhage

Elise N. Erickson [1,2 ✉], Leslie Myatt[1], Joshua S. Danoff [3], Kathleen M. Krol[3] & Jessica J. Connelly[3]

## Abstract

**Background** The oxytocin receptor gene (*OXTR*) is regulated, in part, by DNA methylation. This mechanism has implications for uterine contractility during labor and for prevention or treatment of postpartum hemorrhage, an important contributor to global maternal morbidity and mortality.

**Methods** We measured and compared the level of *OXTR* DNA methylation between matched blood and uterine myometrium to evaluate blood as an indicator of uterine methylation status using targeted pyrosequencing and sites from the Illumina EPIC Array. Next, we tested for *OXTR* DNA methylation differences in blood between individuals who experienced a postpartum hemorrhage arising from uterine atony and matched controls following vaginal birth. Bivariate statistical tests, generalized linear modeling and Poisson regression were used in the analyses.

**Results** Here we show a significant positive correlation between blood and uterine DNA methylation levels at several *OXTR* loci. Females with higher *OXTR* DNA methylation in blood had required significantly more exogenous oxytocin during parturition. With higher DNA methylation, those who had oxytocin administered during labor had significantly greater relative risk for postpartum hemorrhage (IRR 2.95, 95% CI 1.53–5.71).

**Conclusions** We provide evidence that epigenetic variability in *OXTR* is associated with the amount of oxytocin administered during parturition and moderates subsequent postpartum hemorrhage. Methylation can be measured using a peripheral tissue, suggesting potential use in identifying individuals susceptible to postpartum hemorrhage. Future studies are needed to quantify myometrial gene expression in connection with *OXTR* methylation.

## Plain language summary

Oxytocin is a hormone produced by the body during childbirth and can cause contractions of the uterus (womb). Synthetic oxytocin is used as a medicine for stimulating or increasing uterine contractions and controlling bleeding after birth. The oxytocin receptor gene, which enables the body to use oxytocin, can be altered by a chemical modification called DNA methylation. We found that the those who bled more during childbirth had higher oxytocin receptor gene DNA methylation compared to those who had normal bleeding. Higher methylation was also linked to needing greater amounts of oxytocin during labor to achieve vaginal birth and control bleeding. These findings identify that certain problems during birth may be related to oxytocin receptor gene methylation. This research could lead to improvements in how versions of oxytocin are used during the birth process by using the amount of oxytocin receptor gene methylation to predict people who may have problems with uterine contractions or bleeding.

[1]Oregon Health and Science University, Portland, OR, USA. [2]University of Arizona, Tucson, AZ, USA. [3]University of Virginia, Charlottesville, VA, USA.
✉email: eliseerickson@arizona.edu

Successful physiologic parturition resulting in vaginal birth of an infant, placenta and survival of the mother depends upon effective uterine contractility[1]. The myometrium of the uterus consists of layers of smooth muscle that utilize several proteins to generate coordinated and rhythmic contractions that may last from hours to days throughout the process of parturition. These proteins include oxytocin receptor (OXTR), gap junction (Connexin-43) and several isoforms of prostaglandin receptor[2–4]. While the action of prostaglandins primarily prompt labor onset, the latter stages of labor (active and expulsive/second stage and placental delivery) are more dependent upon oxytocin, as a potent uterotonic hormone[5–8].

Oxytocin produced by the placenta can stimulate some uterine contractility[9,10]. However, in advanced stages of labor, a positive feedback mechanism, resulting from vaginal distension and the descending fetal head/body, triggers maternal hypothalamic oxytocin pulsation, and facilitates the final stage of labor[11,12]. Following birth of the newborn, maternal oxytocin continues to stimulate uterine contraction, causing placental detachment from the uterine wall and expelling it from the body[13–15]. Contractions during the third stage of labor also minimize bleeding at the placental site as the blood vessels that were supplying the placenta close off. Clotting factors aid in uterine tone and provide stasis to the open vessels[16]. While some bleeding is normative or physiologic after birth[17,18], if uterine contractions are delayed, infrequent or absent (uterine atony), rapid and excessive bleeding after birth can occur.

Oxytocin acts upon OXTR expressed in smooth muscle cells within the myometrium[4]. OXTR is also present in many regions of the central nervous system[19] and other peripheral organs/tissues[20]. The regulation of OXTR in the myometrium has been investigated for its role in labor processes[4] with the quantity and responsiveness of OXTR playing an important role in uterine contraction during labor[21]. Furthermore, variability in OXTR function could also influence the postpartum outcomes of uterine atony and subsequent postpartum hemorrhage[22].

Postpartum hemorrhage is currently defined as blood loss accompanied by signs of hypovolemia or an accumulation of 1000 mL or greater blood loss in the first 24 h postpartum[23]. Earlier definitions classified postpartum hemorrhage as 500 mL after vaginal birth and 1000 mL after Cesarean birth. Hemorrhage is a leading cause of maternal death worldwide[24], and though it accounts for only 11.4% of maternal mortality in the United States[25], recent data indicate that postpartum hemorrhage itself and/or the need for life-saving interventions (blood transfusions, surgical interventions) to control or treat postpartum hemorrhage, are on the rise[26–29]. Sequela of postpartum hemorrhage beyond the immediate hospitalization period include prolonged recovery from childbirth, delays in lactation onset or difficulty producing milk[30], anemia[31,32], fatigue, and some studies have linked postpartum hemorrhage with postpartum depression and post-traumatic stress[33–35]. Other labor/maternal characteristics associated with postpartum hemorrhage include length of labor, genital tract lacerations, fetal macrosomia, hypertension/preeclampsia, or a prior history of postpartum hemorrhage[28,36]. Many factors are closely associated with each other, which compounds the risk to the individual and complicates research methods. Regardless of known risk factors, up to 40% of postpartum hemorrhage occurs in individuals without risk factors, making postpartum hemorrhage often difficult to anticipate[27,37–39].

Despite the important role of OXTR during parturition, little has been published about epigenetic regulation of the oxytocin receptor gene, OXTR, in connection with the normal process of parturition including postpartum uterine contractility. Located on chromosome 3, OXTR consists of three introns and four exons[40]. Previous research has shown that DNA methylation of a region contained within a CpG island in the human OXTR promoter (termed MT2) plays a key role in regulating OXTR transcription[41]. In humans and animal (prairie vole) models, our team has shown that both brain and blood-derived DNA methylation in this region is correlated with gene expression in the brain. Blood methylation accounts for approximately 18% of the variance in expression[42], suggesting to some extent that blood-derived DNA methylation measurements in OXTR may be a proxy for gene expression in inaccessible tissues[42,43].

Researchers examining OXTR DNA methylation (OXTRm) have found that various psychopathologies are correlated to OXTRm dysregulation[44–46]. Others have reported variation in social/emotional outcomes in connection with OXTRm[47,48]. We and others have also identified a role for OXTRm in the risk for postpartum depression[44,49], preterm labor[2,50], maternal antidepressant use[51], and exposure to early-life adversity/ posttraumatic stress[52–57]. Autism and social cognition have also been linked to OXTRm[58–63]. Finally, promoter OXTRm has also been found to be sensitive to early caregiving experience, such that higher levels of parental care have been associated with reduced DNA methylation in both human infants[64] and prairie vole pups[42].

Understanding how OXTRm also relates to OXTR function and subsequent uterine contractility during parturition would have important clinical implications both physiologically and pharmacologically. Oxytocin administration is the primary pharmacological intervention used during the course of labor[65,66] and is frequently used for labor induction[67], rates of which have risen from 9.5% of births in 1990[68] to 31.4% of births in 2020[69]. Intrapartum dose requirements for oxytocin have been found to vary by body mass index (BMI)[70,71] and maternal age[72], and may be influenced by differences in lipid metabolism[73]. While use of oxytocin for labor stimulation may be a life-saving intervention, overuse can expose the mother to risks, including postpartum hemorrhage[74–76]. Intrapartum oxytocin use leads to uterine oxytocin receptor desensitization (and internalization of the receptor from the myocyte membrane) and/or downregulation of OXTR[21,22,77,78]. These mechanisms result in diminished capacity of the uterus to contract effectively after birth and predisposes the individual to a higher likelihood of uterine atony and postpartum hemorrhage. Rates of atony-associated postpartum hemorrhage appear to be specifically rising[79], which is tied to increased intrapartum oxytocin administration[80] even among healthy, low-risk vaginal births[81]. Oxytocin also remains a first-line therapy for stimulating uterine contraction after birth for prevention and treatment of postpartum hemorrhage[82].

Other than three preterm labor focused studies already mentioned, no studies have linked OXTRm to maternal oxytocin response or uterine contractility throughout parturition; however, extracting myometrial tissue would be impractical and unethical following vaginal birth. A peripheral tissue (e.g., whole blood) could be used to understand uterine OXTRm patterns if the tissue methylation levels were well-correlated with one another. Our previous work has reported significant correlations between OXTRm derived from brain and blood in prairie voles[42] as well as blood and saliva in humans;[61] however, no studies have reported correlation of blood and myometrial DNA methylation.

In the current study, we address this scientific/clinical gap by examining epigenetic variability in the endogenous oxytocin system. First, we measure OXTRm between matched uterine myometrium and maternal blood samples using an exploratory array method as well as targeted pyrosequencing to determine if blood can be used as an indicator of uterine OXTRm levels. Second, we test differences in OXTRm in relation to oxytocin administration during the process of parturition and indices of postpartum hemorrhage. Considering that postpartum

hemorrhage is commonly attributed to uterine atony, reduced availability of OXTR might associate with greater postpartum hemorrhage risk. We therefore hypothesize that higher levels of OXTRm, reflective of less OXTR availability, might associate with greater blood loss after giving birth. In this study, we find that OXTRm in myometrium correlates with the level of DNA methylation from blood and that higher OXTRm relates to both greater oxytocin dosage needed during the birth process and more blood loss after vaginal birth. Findings from this study may further our understanding of the relationship between the endogenous oxytocin system and birth outcomes, inform clinical risk assessment, or help improve use of oxytocin administration.

## Materials and methods
### Correlation between OXTRm using matched uterine myometrium and maternal blood
*Participants*. Participants' data in this study were available via a pregnancy-related tissue repository managed by Oregon Health and Science University. Potential participants were engaged by trained research staff, and after providing written informed consent, donated samples of myometrium, whole maternal blood, placenta and cord blood. The pregnancy repository was approved by the ethics committee at Oregon Health and Science University. Limited clinical information was available in the repository and no identifying information, therefore the committee found it did not constitute research with human subjects. Samples available in the repository were from participants who gave birth at two different institutions, (Oregon Health and Science University and Chelsea and Westminster Hospital, London). Material data transfer and data use agreements were established between institutions, de-identified samples were collected from (Chelsea and Westminster Hospital) and included in an existing tissue repository at (Oregon Health and Science University).

### DNA obtained from whole blood and myometrium at time of planned cesarean delivery
Delivery at 37 weeks or greater was indicated due to prior cesarean, choice, or presence of chronic or gestational hypertension. In the hour prior to delivery, whole blood was collected in standard EDTA tubes and frozen in 1.5 mL aliquots at −80 °C. After birth of the newborn and placenta, a small portion of full-thickness myometrium (measuring approximately 2 cm × 2 cm in size) was excised from the hysterotomy incision and was handed off the sterile field to study staff, described by McElvey et al. (2000)[83]. Myometrium was rinsed in phosphate buffered saline, dissected from serosa and endometrium, if present, and flash frozen in liquid nitrogen in small sections. Uterine specimens were not collected in an RNA preservation medium. Specimens were collected from 27 participants. DNA from blood and myometrium were isolated, bisulfite converted and followed by both Illumina array processing and pyrosequencing to assess the methylation across the OXTR as well at the MT2 locus of interest (−934) (complete methods described below).

### Blood-derived OXTRm in relationship to obstetric oxytocin requirements during parturition and postpartum hemorrhage (case-control study)
Postpartum hemorrhage cases were matched to participants with normal/typical blood loss after birth. Participants were matched by induction of labor status (vs. spontaneous labor onset) and parity. The protocol for this study underwent ethics review at (Oregon Health and Science University) and was approved as human research.

### Participants
Participants gave birth in the greater Portland, Oregon area including hospitals and community birth settings (home/birth center) from November 2018 through February 2020. Importantly, study recruitment and planned study visits for blood sampling were halted prematurely due to SARS-CoV-2 pandemic restrictions on in-person research activities. Those giving birth at our institution were given information about the study in the postpartum unit or at a follow-up visit based on meeting basic screening criteria in the electronic health record. Other participants responded to advertisements posted in the community and on social media.

Enrollment criteria included the following: singleton vaginal birth, 37 weeks or greater gestational age, maternal age 15–45, able to read and provide informed assent/consent in English or Spanish. Exclusions included: known coagulopathy, hemolysis, elevated liver enzymes and low platelet (HELLP) syndrome, disseminated intravascular coagulation, magnesium sulfate administered during labor, placenta accreta, vasa previa, gestational thrombocytopenia (< 80,000 platelets/mL) or severe bleeding noted from genital tract injury during birth. Participants completed baseline survey measures upon enrollment after a process of informed written consent was conducted. Recruitment goal of 100 cases and 100 matched controls was based on a mean difference between groups of a magnitude of two percentage points (SD of 5.0) with 80% power and α of 0.05. Of the 577 postpartum records screened, we approached 393 individuals and 120 met criteria and consented to participate in the study.

### Cases of postpartum hemorrhage and characteristics of uterine atony
We enrolled participants as cases who had experienced a diagnostic postpartum hemorrhage, a cumulative blood loss ≥ 1000 mL in the first two hours after birth, when bleeding was attributed to lack of uterine tone. People with significant bleeding from lacerations were excluded. We also enrolled those who experienced uterine atony with blood loss totaling at least 400 mL in the first two hours after birth and had additional clinical interventions including secondary uterotonic medication to treat heavier than normal bleeding (not attributed to lacerations). These interventions may have led to avoiding an official postpartum hemorrhage diagnosis. Further rationale for this expanded definition of postpartum hemorrhage is supported by other research indicating that providers commonly underestimate the volume of blood lost during birth[84,85].

This phenotyping process required extensive review of the electronic medical record for nurses' notes and the birth attendant's (midwife/physician) written birth narrative, in addition to viewing the medication administration records and postpartum notes. Care providers reported blood loss in the health record as either measured (e.g., blood loss was weighed or measured using graduated drapes) or by estimation methods (visual inspection of drapes and towels). We also examined post-birth hemoglobin/hematocrit values to validate the quantity of blood lost, when it was available in the chart.

### Medical record abstraction and oxytocin administration quantification
The electronic medical record was abstracted for the dose, start and stop times of any oxytocin administered during labor, along with the timing and duration of titration of oxytocin doses. The dosing of oxytocin is quantified in milliunits (mU)/min; therefore, the total cumulative dosage was a product of the duration (min) of a given level of titration and the oxytocin dose (i.e., 60 min x 2 mU/min = 120 mU). Highest dose given along with duration of oxytocin, other medications given during labor and any prenatal, intrapartum or postpartum complications was also abstracted. Normal or healthy labors will not typically require augmentation with oxytocin, or oxytocin will be used for only a short time, for example, following regional analgesia

(epidural) placement. In the presence of labor induction or need for labor augmentation, oxytocin regimens typically involve starting at a 1 mU/min administration rate and increasing by 1–2 mU/min every 20–40 min[86]. Significant institutional variation as well as situational variability in dosing exists, however.

**Treatments and interventions to control atony and postpartum hemorrhage**. Similarly, oxytocin given after birth for postpartum hemorrhage prophylaxis (active management of the third stage of labor) was recorded in addition to treatments for heavier bleeding including any of the following: intravenous (IV) oxytocin, intramuscular oxytocin, misoprostol, methylergonovine, carboprost tromethamine, bimanual compression, uterine massage, bladder management, manual removal of retained clots or placental/membrane fragments. Routine active management of third stage labor may consist of 10 U of oxytocin given via intramuscular injection or through the intravenous (IV) line. However, some institutions will instead utilize the existing IV solution of oxytocin being used for labor management, which may be 30 U oxytocin in 500 mL lactated ringers (however, some institutions may use 40 U in 1000 mL)[66]. If bleeding is ongoing, an IV solution of oxytocin would likely be ordered and administered rapidly. The medical record was examined for the dose/duration of the postpartum infusion (if any was used) as well as any intramuscular doses. Some births did not have any postpartum oxytocin administration, which may occur if a patient declined the medication and/or the practitioner assessed the patient to be low-risk for postpartum hemorrhage. In addition, the postpartum records were reviewed for IV iron therapy or blood transfusions as well as delayed postpartum hemorrhage (> 2 h after birth). In the enrollment survey, participants were asked to self-report their pregnancy, birth and medical history, which was used to confirm and validate medical record abstraction findings.

**Case-control maternal blood-derived DNA sampled 6–10 weeks after birth**. Timing of blood sampling was delayed until 6–10 weeks after birth to limit chimerism with any DNA/blood cells received via blood transfusion among participants with postpartum hemorrhage. Whole blood samples were obtained from participants using vacutainer EDTA tubes. Samples were stored immediately at −80 °C. Participants received remuneration in the form of a $20 gift card at the time of the study visit. A total of 91 blood samples were available for DNA extraction. Several participants ($n = 6$) could not have blood sampling due to pandemic restrictions and another 23 enrolled participants did not show for the study visit and could not be contacted for rescheduling. DNA isolated for pyrosequencing occurred using reagents and procedures outlined in the QIAamp DNA Mini Kit (Qiagen, Hilden, Germany). All DNA was quantified using Nanodrop. One-hundred nanograms (ng) of isolated DNA were subject to bisulfite conversion using MECOV50 kits (Thermo Fisher Scientific, Waltham, USA). Sample locations within a 96-well plate were determined using an online random number generator (random.org).

**Determination of targeted CpG methylation by pyrosequencing**. Following bisulfite conversion, the remaining procedures were performed in triplicate: Forty nanograms of bisulfite-converted DNA was amplified using polymerase chain reaction (PCR) with PyroMark PCR Kits (Qiagen) and 0.2 μM primers (Forward (TSL101F): 5′-TTGAGTTTTGGATTTAGATAATTAAGGATT-3′; Reverse (TSL101R): 5′-biotin-AATAAAATACCTCCCACTCCTTATTCCTAA-3′). Methylation standards (0, known, and 100% methylated) and negative controls from bisulfite conversion and PCR were included on the PCR plate. The following cycling protocol occurred simultaneously on three

identical satellite thermocyclers (Bio-Rad, Hercules, USA): Step 1) 95 °C, 15 min; Step 2) 50 cycles of: 94 °C, 30 s; 56 °C, 30 s; 72 °C, 30 s; Step 3) 72 °C, 10 min; Step 4) 4 °C hold. The amplification of a 116-base-pair region on the coding strand of *OXTR* containing CpG −934 and −924 (hg38, chr3: 8,769,044 – 8,769,160) and no contamination in negative controls was confirmed by 2% agarose gel electrophoresis.

DNA methylation level for each sample was assessed through pyrosequencing using PyroMark Gold Q24 reagents on a PyroMark Q24 machine (Qiagen) (sequencing primer TSL101S: 5′-AGAAGTTATTTTATAATTTTT-3′). Samples with high replicate variability were identified as having a Z-score above +/−1.96. Eight samples were identified on the basis of this criterion. These 8 samples were visually inspected; when one replicate value significantly deviated from the other two values, this value was removed ($n = 7$). Replicate variability, assessed via mean deviation, averaged +/−1.33% for the tissue match experiment (blood = +/−1.33%, tissue = +/−1.33%) and +/−1.29% in the case-control study. Apart from samples reporting on two replicate values, reported epigenotypes are the average of three replicate values. In addition, 0% and 100% DNA methylation controls as well as a sample of known DNA methylation level, were run alongside experimental samples with expected methylation results (0% *OXTR*m mean = 3.57% +/− 0.93%, 100% *OXTR*m mean = 99.06% +/− 0.78%, and known 42.5% +/− 1.88%).

**Illumina Array (myometrial tissue and matched blood only)**. Isolated DNA was quantified by fluorescence using Quant-iT™ PicoGreen ® dsDNA assay kit (Invitrogen | Thermo Fisher Scientific). DNA methylation was assessed using Illumina's Methylation EPIC BeadChip (Illumina, San Diego, USA). 500 ng of each sample was bisulfite converted using an EZ DNA Methylation Kit (Zymo), amplified, hybridized, and imaged. DNA methylation data for over 850,000 CpGs was generated per sample and preprocessed using R statistical suite (version 3.6.1).

Raw.idat files were read and preprocessed using the *minfi* R package[87,88]. The data set was preprocessed using noob for background subtraction and dye-bias normalization. All methylation values with detection $P > 0.01$ were set to missing. One uterine sample had excessive missing probes (809,422, 93.44%) and was excluded from further analysis. Of the remaining samples, the number of missing probes ranged from 847 to 5008 (median: 1106). Probes with > 1% missing values ($n = 12,399$) were removed from further analysis. Additionally, samples were checked for unusual cell mixture estimates using the using the Houseman method as implemented in minfi[89,90]. All samples passed these quality controls. Principal components analysis, as implemented in the shinyMethyl R package, was used to examine batch effects[91]. The first seven principal components were examined using plots and potential batch effects were tested using linear models. Principal component 1, which accounted for 79.47% of the total variance, was associated with tissue type ($F_{(1, 50)} = 5149$, $p < 2.2e-16$, adjusted $R^2 = 0.992$). Location of specimen collection site (London, UK or Portland, OR) was associated with principal components 2, 3, 6, and 7, which account for 1.57%, 0.95%, 0.72%, and 0.69% of the total variance, respectively (PC2: $F_{(1, 50)} = 4.34$, $p = 0.04$, adjusted $R^2 = 0.061$; PC3: $F_{(1, 50)} = 4.57$, $p = 0.038$, adjusted $R^2 = 0.066$; PC6: $F_{(1, 50)} = 6.48$, $p = 0.014$, adjusted $R^2 = 0.097$; PC7: $F_{(1, 50)} = 16.04$, $p = 2.06e-4$, adjusted $R^2 = 0.228$). Array was associated with PC2 ($F_{(6, 45)} = 30.97$, $p = 2.01e-14$, adjusted $R^2 = 0.779$). Position on the array was not associated with any of the first seven principal components. Because proportions of variance explained by associated principal components were low, no batch correction

method was used. The ewastools R package was used to assess Illumina quality control metrics and call genotypes and donor IDs to ensure the identity of matched samples from the same individual[92]. One uterine sample failed the non-polymorphic green control, but was retained as it was just below threshold and did not appear abnormal on the other 16 metrics. All other samples passed all Illumina quality controls. Following quality controls, background subtraction and normalization were performed using the preprocess Funnorm function in the *minfi* package, which is recommended for between-tissue studies[93]. Normalized beta values were then extracted for probes that fall within *OXTR* and probes within 3 kb of the gene (hg19 chr3: 8,789,094-8,814,314). There are 23 such probes, but one probe (cg08449558) was excluded because it includes a SNP with minor allele frequency of 9%.

**Statistics and reproducibility**. DNA methylation levels between matched blood and uterine specimens were assessed with a Spearman correlation. Paired t-tests (two-sided) were used to compare mean differences in level of methylation observed in each tissue. To correct for multiple comparisons, we applied a Benjamini-Hochberg correction to the *p*-values from the correlation analysis.

For the case-control study, differences in sample characteristics were measured between cases and controls using parametric and non-parametric tests as appropriate (Spearman, two-sided t-tests or $\chi^2$). We also examined differences in *OXTR*m by baseline characteristics (e.g., age, body mass index, parity). Our primary hypothesis was that higher *OXTR*m would be associated with higher blood loss and postpartum hemorrhage occurrence and that this relationship would be moderated by oxytocin use during labor. We therefore tested this hypothesis using an interaction term of oxytocin use (versus no oxytocin use) in the regression models.

The intent for this study was a 1:1 match of hemorrhage cases with control (typical postpartum bleeding) participants. However, in the end, cases outnumbered controls as recruitment and study visits stopped in February of 2020. We therefore analyzed our data comparing cases to control participants in addition to using the volume of blood loss as a continuous outcome. First, we compared DNA methylation between cases and control groups with linear regression. Controlling for mean replicate deviation, we then examined this association with or without oxytocin use in labor. We then used generalized linear models with a gamma distribution to examine the outcome of the quantity of oxytocin used across the phases of birth (oxytocin quantity was significantly right skewed) (intrapartum and postpartum) with *OXTR*m as the primary independent variable controlling for mean deviation (across replicates). We included parity and body mass index at birth as confounders in the models as they significantly differed between cases and controls and both are associated with oxytocin use during childbirth.

In the second model, we used adjusted Poisson regression with robust standard errors to assess the relationship of *OXTR*m to the case or control status. The third model, a multivariable generalized linear model (GLM), assessed the total blood lost during the birth process using a gamma distribution (blood loss is skewed). The fourth and fifth models used adjusted Poisson regression with robust standard errors for the outcomes of postpartum hemorrhage (using the 500 mL definition) and again at the 1000 mL definition. For each of these models we included the interaction term of oxytocin use during labor (versus no oxytocin) and controlled for replicate variability, parity, antibiotic use during labor (to control for possibility of intrapartum infection) and body mass index at delivery. Both the traditional

and updated definitions of postpartum hemorrhage were used, given the known problems with blood loss underestimation[94].

**Reporting summary**. Further information on research design is available in the Nature Portfolio Reporting Summary linked to this article.

## Results

**Uterine myometrium and maternal blood are correlated at multiple CpG sites in *OXTR***. Matched blood and myometrium samples from 26 participants were included in the analysis. The mean (standard deviation) age of the sample was 35.3 (5.9) years. Participants delivered via planned, pre-labor Cesarean with a mean (standard deviation) gestational age of 38.7 (0.77) weeks. The majority were multiparous (85%) with a mean (standard deviation) pre-pregnancy BMI of 27.8 (7.8) kg/m².

We analyzed myometrium and blood-derived DNA methylation at CpG site sites across the *OXTR* locus including two previously reported and conserved sites, −924 and −934, located in the promoter region of *OXTR* (Fig. 1). Data were generated using array-based methods and targeted pyrosequencing.

Using data from the Illumina methylome array, 22 probes within *OXTR* and its promoter were examined for correlation between maternal blood and myometrium using Spearman's rho and Benjamini-Hochberg correction (Fig. 1 denotes specific loci and Fig. 2 shows significant sites). Of these 22, four sites in the 3' region of the gene were significantly correlated between tissues cg03257388 ($\rho(24) = 0.66$, $p = 0.005$), cg02192228 ($\rho(24) = 0.62$, $p = 0.008$), cg11171527 ($\rho(24) = 0.55$, $p = 0.02$) and cg15317815 ($\rho(24) = 0.59$, $p = 0.01$). In addition, two CpG sites not present on the array, but shown to partially account for gene expression and highly reported on in the literature, were assayed by pyrosequencing and found to be correlated (−934: $\rho(24) = 0.65$, $p = 0.005$) (−924: $\rho(24) = 0.52$, $p = 0.03$). Given that site −934 (chr3: 8,810,729-8,810,845) demonstrated both high statistical significance between tissues, is associated with differences in transcription of *OXTR*, and is a commonly studied CpG site in human adult literature[43,95], we chose to focus on this site for our examination of oxytocin administration and postpartum hemorrhage outcomes in the case-control study.

**Postpartum hemorrhage cases were administered higher dosage of oxytocin used throughout labor and postpartum**. Using a case-control study design, we enrolled a sample of 69 postpartum hemorrhage cases and 50 control participants. Blood samples were available for DNA isolation from 55 of the cases and 36 of the control participants. Table 1 denotes sample characteristics and differences between cases and controls. While we did match for parity during enrollment, nulliparous participants were more highly represented among cases compared to controls (57.9% vs 38.0%, $p < 0.05$) due to halted enrollment. The range of total blood loss was 400–2540 mL among cases and 50–600 mL for control participants. No differences were seen between cases and controls in maternal age, gestational age, and presence of antenatal complications (hypertension/diabetes). Participants self-identifying as Latin American / Hispanic ethnic background/ancestry were more frequent among the cases ($n = 11,15\%$ of cases, vs. $n = 1$, 2% of controls, $p < 0.05$). Other differences included use of antibiotics during labor and BMI, with postpartum hemorrhage cases having significantly higher BMI at delivery (31.9 vs. 29.9, $p = 0.03$). Finally, the total quantity, duration and maximum dosage of oxytocin needed during labor was higher among cases of postpartum hemorrhage. As expected, postpartum use of

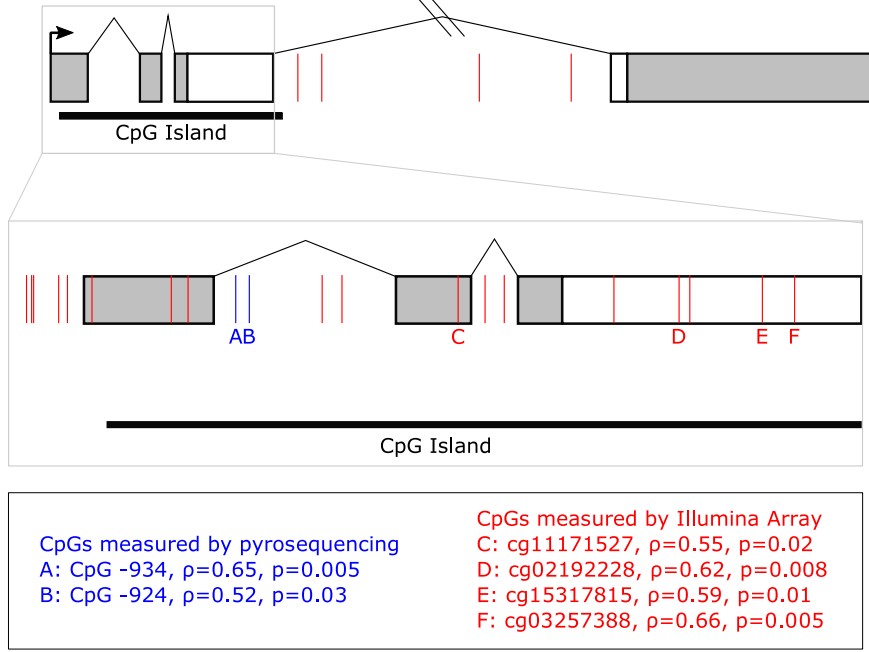

**Fig. 1 OXTR gene schematic and location of CpG sites with significantly correlated methylation.** Gene schematic of *OXTR*. Boxes represent exons and lines represent introns. Coding regions are in white and untranslated regions are in gray. The black arrow indicates the transcription start site. The black bar below the gene denotes the CpG island. The 5′ region of the gene is enlarged to show detail. Blue lines indicate CpG sites assayed by pyrosequencing and red lines indicate CpG sites assayed by Illumina Methylation EPIC 850 K array. CpG sites with significantly correlated DNA methylation in blood and myometrium are indicated by letters and Spearman's rho is reported below the gene schematic.

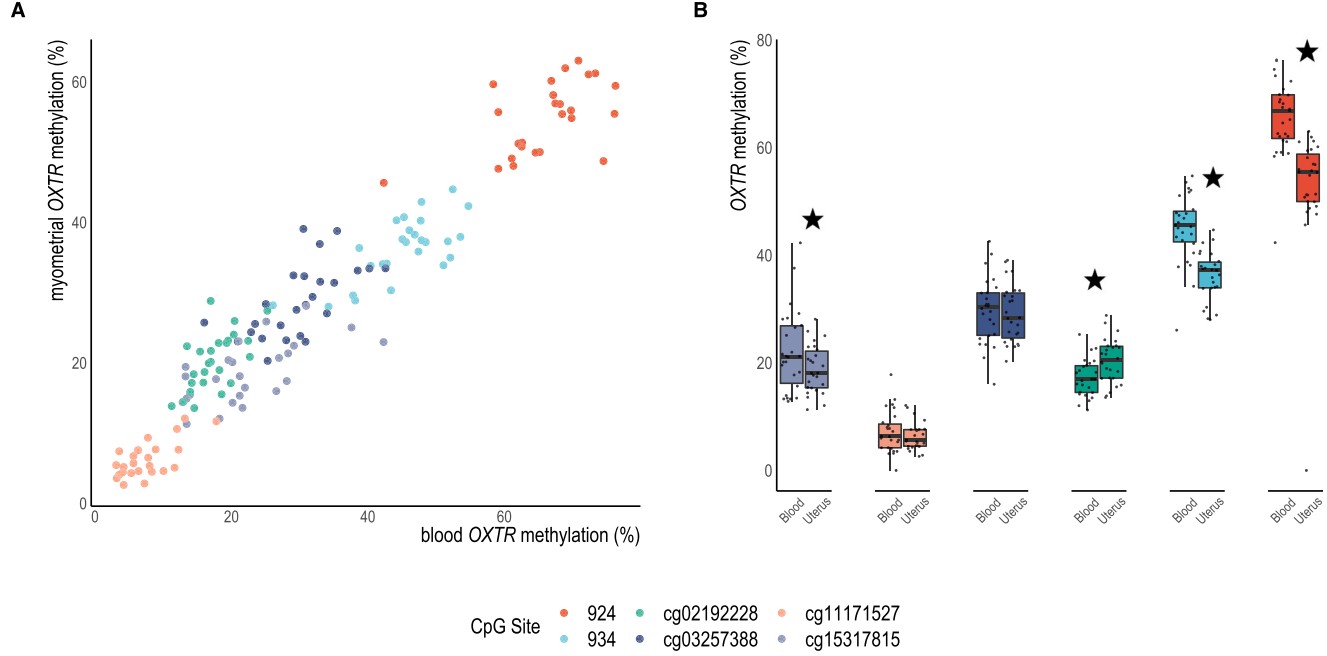

**Fig. 2 DNA methylation of CpG sites in OXTR are correlated in blood and myometrium. A** Significantly correlated CpGs between tissues ($n = 26$ participants with matched samples) $p < 0.05$ after correction for multiple comparisons using the Benjamini-Hochberg method: −924 (red): $\rho = 0.52$, $p = 0.03$, −934 (light blue): $\rho = 0.65$, $p = 0.005$, cg02192228 (green): $\rho = 0.62$, $p = 0.001$, cg03257388 (dark blue): $\rho = 0.66$, $p = 0.005$, cg11171527 (apricot): $\rho = 0.55$, $p = 0.02$, cg15317815 (lavender): $\rho = 0.59$, $p = 0.01$. **B** Differences in methylation levels at each CpG site between myometrium and blood, stars denote Bonferroni corrected significant differences: −934 ($t(25) = 10.1$, $p = <0.001$), −924 ($t(25) = 9.6$, $p < 0.001$), cg15317815 ($t(24) = 3.2$, $p = 0.004$), cg02192228 ($t(24) = -4.3$, $p = 0.003$), cg03257388 ($t(24) = 1.12$, $p = 0.27$), cg11171527 ($t(24) = 2.2$, $p = 0.04$). Whiskers denote $+/-1.5$ x interquartile range.

**Table 1 Significant differences in total oxytocin administration during labor between postpartum hemorrhage cases compared to controls, matched on parity and labor induction and other sample characteristics.**

| | Case ($n = 69$) Mean (SD) | Control ($n = 50$) Mean (SD) |
|---|---|---|
| Maternal age, years | 31.4 (4.9) | 32.4 (4.4) |
| Body Mass Index at delivery, kg/m$^2$ | 31.9 (5.4) | 29.9 (4.2) * |
| Gestational age, weeks | 39.7 (1.3) | 39.8 (1.2) |
| Third trimester hemoglobin mg/dl | 11.8 (1.2) | 11.9 (1.2) |
| Length of first stage labor, hours | 16.4 (11.3) | 11.6 (7.8) * |
| Length of second stage labor, hours | 1.6 (1.6) | 1.3 (1.5) |
| Length of ruptured membranes, hours | 9.7 (10.9) | 6.7 (6.8) |
| Total oxytocin if administered intrapartum, Units | 9.9 (9.4) | 3.9 (4.7) † |
| Total oxytocin duration intrapartum, hours | 16.3 (9.7) | 10.5 (7.5) † |
| Highest maximum dose of oxytocin, mU/min | 13.9 (8.1) | 9.1 (6.1) † |
| Postpartum oxytocin used, Units | 21.5 (14.6) | 12.5 (9.1) † |
| Total oxytocin used intrapartum/postpartum combined, Units | 25.6 (16.2) | 12.4 (9.4) ‡ |
| Infant size, grams | 3410.7 (568.5) | 3484.6 (689.9) |
| Length of third stage of labor, minutes | 10.0 (9.5) | 9.0 (10.4) |
| Total volume blood loss, mL | 813.9 (511.9) | 277.1 (142.8) ‡ |
| Total volume blood loss, range | 400–2540 | 50–600 |
| Difference in hemoglobin (third trimester to lowest postpartum value), mean (SD) mg/dl | −1.9 (2.3) | 0.05 (1.23) ‡ |
| | Case ($n = 69$) n (%) | Control ($n = 50$) n (%) |
| Primiparous | 40 (57.9) | 19 (38.0) * |
| Self-reported ancestry/race/ethnicity[a] | | |
|    European | 55 (79.7) | 39 (78.0) |
|    Latin American | 11 (15.9) | 1 (2.0) * |
|    Asian | 7 (10.1) | 6 (12.0) |
|    African | 1 (1.4) | 0 (0.0) |
|    Other | 4 (5.8) | 3 (6.0) |
| Bleeding during first trimester pregnancy | 7 (10.0) | 1 (2.0) |
| Prior Cesarean birth | 6 (8.8) | 4 (8.0) |
| Tobacco used prior to/during pregnancy | 4 (5.8) | 7 (14.0) |
| Gestational diabetes | 13 (18.5) | 5 (10.0) |
| Mild gestational thrombocytopenia (platelets 80,000–120,000/µl prior to delivery) | 4 (5.8) | 2 (4.0) |
| Hypertensive disorder diagnosis | 6 (8.7) | 4 (8.0) |
| Location of birth | | |
|    Hospital | 68 (97.1) | 47 (95.9) |
|    Community (home/birth center) | 2 (2.9) | 2 (4.0) |
| Group B streptococcus carrier | 22 (31.9) | 8 (17.0) |
| Infection/ chorioamnionitis suspected during labor | 8 (11.6) | 2 (4.0) |
| Antibiotics administered | 30 (43.5) | 9 (18.8) † |
| Epidural use in labor | 49 (71.0) | 30 (61.2) |
| Spontaneous onset labor | 34 (49.3) | 26 (52.0) |
| Oxytocin used in labor | 42 (60.9) | 28 (56.0) |
| Instrument assisted vaginal birth | 5 (7.2) | 1 (2.1) |
| Genital trauma (none/no repair needed) | 17 (24.3) | 14 (29.8) |
| Female infant | 37 (53.6) | 21 (42.0) |
| Active management of third stage labor (oxytocin) | 58 (84.1) | 41 (82.0) |
| Timing of cord clamping < 30 seconds | 8 (12.5) | 3 (6.9) |
| Received iron infusion or blood transfusion postpartum | 13 (18.8) | 1 (2.1) † |
| Available DNA specimens | 55 (80.0) | 36 (71.4) |

* $p < 0.05$
† $p < 0.01$
‡ $p < 0.001$
[a]May have selected one or more ancestry/ethnicity.

oxytocin was also significantly higher among cases, though the use of prophylactic oxytocin (active management of third stage labor) was not different between groups. The cumulative oxytocin dosage for cases was more than double that of controls (25.6 Units vs. 12.4 Units, $p < 0.001$).

**Maternal *OXTR*m was lower among cases of postpartum hemorrhage following labor without oxytocin.** Average *OXTR*m across the 91 participants was 45.2% (SD = 6.5), ranging from 20.3% to 63.2%. Mean (SD) % *OXTR*m for cases was 44.9 (7.2) and 45.4 (5.4) for control participants, which was not significantly different (z = −0.02, $p = 0.99$, Wilcoxon rank-sum). Figure 3

illustrates the DNA methylation differences between cases and controls measured by Wilcoxon rank-sum tests when oxytocin was used in labor ($n = 53$) z = −1.86, $p = 0.06$, or when labor progressed without oxytocin stimulation ($n = 38$) z = 2.84, $p = 0.003$. Using a linear regression model and controlling for pyrosequencing replicate variability, we found *OXTR*m to be significantly lower in cases compared to controls ($n = 22$ cases and 16 controls) when labor occurred without oxytocin stimulation (4.15% lower; 95% CI −7.91 to −0.39) (Table 2). Parity, maternal age, gestational age, and maternal BMI at delivery were not associated with *OXTR*m in bivariate analyses ($p$-values > 0.05).

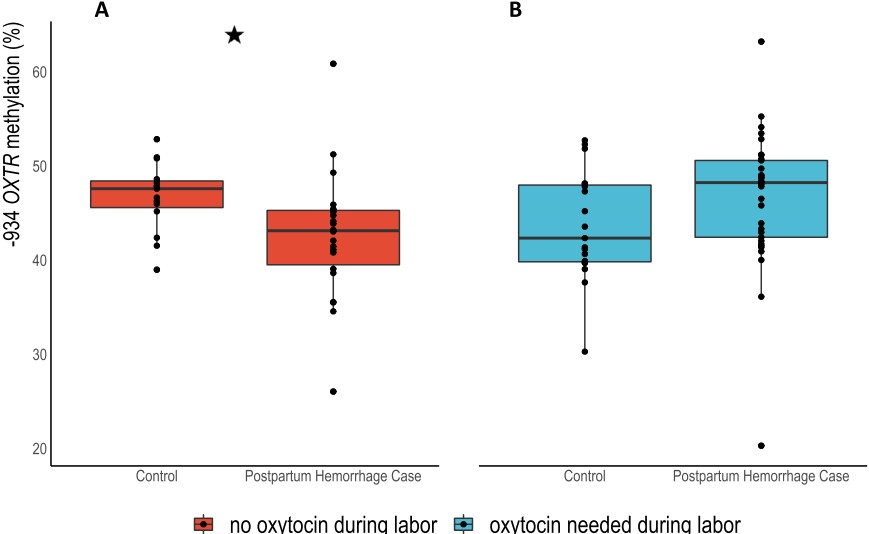

**Fig. 3 Cases of postpartum hemorrhage had lower DNA methylation when labor was not stimulated by oxytocin. A** Star denotes significant difference in total DNA methylation between cases of postpartum hemorrhage cases and controls among participants whose labor was not stimulated with oxytocin (red box plots, $n = 38$), $z = 2.84$, $p = 0.003$. **B** Difference in DNA methylation between postpartum hemorrhage cases and controls when oxytocin was used in labor (blue box plots, $n = 53$) $z = -1.86$, $p = 0.06$. Whiskers denote $+/-1.5$ x interquartile range.

**Table 2 Linear regression with interactions demonstrates higher *OXTR*m in postpartum hemorrhage cases when oxytocin was administered during labor and lower *OXTR*m when oxytocin was not needed in labor.**

|  | Cases vs. Controls $\beta$(95% CI) |
| --- | --- |
| *OXTR*m |  |
| Sample without interaction ($n = 91$) | −0.50 (−3.27 to 2.28) |
| Oxytocin during labor ($n = 53$) | 2.16 (−1.76 to 6.09) |
| Without oxytocin in labor ($n = 38$) | −4.15 (−7.91 to −0.39) * |
| Interaction: Oxytocin in labor x Case vs. Control ($n = 91$) | 6.29 (0.80 to 11.79) * |
|  | Sample Characteristics $\beta$(95% CI) |
| *OXTR*m |  |
| Maternal age | 0.08 (−0.23 to 0.39) |
| Parity | 0.07 (−1.10 to 1.24) |
| Body Mass Index | 0.09 (−0.18 to 0.37) |
| Gestational Age | −0.52 (−1.64 to 0.61) |
| Infant sex | −0.17 (−2.90 to 2.56) |
| Smoking before/during pregnancy | 0.71 (−4.10 to 5.50) |
| Antepartum anemia | −0.49 (−3.73 to 2.74) |

*$p < 0.05$
Regression models controlled for replicate variability.
*OXTR*m was not related to other sample characteristics.

**Table 3 Greater total oxytocin administration associated with higher *OXTR*m among individuals who needed oxytocin for labor augmentation or induction of labor, parity and body mass also associated with oxytocin dosage in the multivariable model.**

| Total oxytocin administered (Units) | $\beta$(95% CI) |
| --- | --- |
| *OXTR*m (5% increase) | 2.47 (0.66 to 4.28) † |
| Replicate variability | −0.98 (−30.69 to 28.73) |
| Parity (total prior births) | −2.64 (−4.67 to −0.59) * |
| BMI at delivery | 0.82 (0.14 to 1.51) * |

*$p < 0.05$
†$p < 0.01$

treating postpartum bleeding) ($\beta = 2.47$, 95% CI 0.66-4.28, $p = 0.008$). We controlled for pyrosequencing replicate variability, parity, and maternal BMI at time of delivery in the model (Table 3, Fig. 4).

**Higher *OXTR*m is associated with greater relative risk for postpartum hemorrhage when oxytocin was used during labor.** Examining only participants with oxytocin use during labor, we found higher *OXTR*m was associated with greater blood loss, $\rho(51) = 0.31$, $p = 0.02$. However, among those labors not stimulated with oxytocin, higher *OXTR*m correlated with lower blood loss ($\rho(36) = -0.34$, $p = 0.04$) (Fig. 4). Given that oxytocin use appeared to be an effect modifier, we used Poisson regression models and GLMs with interactions across the entire sample. In the regression models, the association between *OXTR*m and blood loss remained across the hemorrhage outcomes. When oxytocin was used during labor, each 5-percentage point increase in *OXTR*m level was associated with 45% higher relative risk of being a case compared to a control (IRR 1.45, 95% CI = 1.11–1.91). For each 5-percentage point increase in *OXTR*m level, 185.22 mL greater blood loss was predicted in the GLM ($\beta = 185.22$, 95% CI 70.53-299.91). Increased *OXTR*m was also associated with higher relative risk for postpartum bleeding of 500 mL or greater, RR 1.45 (95% CI 1.12–1.89). Finally, the combination of higher *OXTR*m and oxytocin use in labor was

**Oxytocin administration during parturition is associated with greater *OXTR*m.** Cumulative oxytocin administration (intrapartum and postpartum total in Units) was correlated to higher *OXTR*m ($\rho(87) = 0.28$, $p = 0.008$). Among participants who needed oxytocin administered *during* labor, the correlation between total dosage and *OXTR*m was higher ($\rho(50) = 0.42$, $p = 0.002$). As the distribution of oxytocin use in labor was significantly right-skewed (Shapiro-Wilk $W$ test for normality $W = 0.87$, $p < 0.0001$), we used a GLM with a gamma distribution to examine oxytocin use by *OXTR*m in lieu of a linear regression or transforming the data. We found that with an increase in *OXTR*m of 5 percentage points, 2.48 more units of oxytocin were required for management of parturition (stimulation of labor and

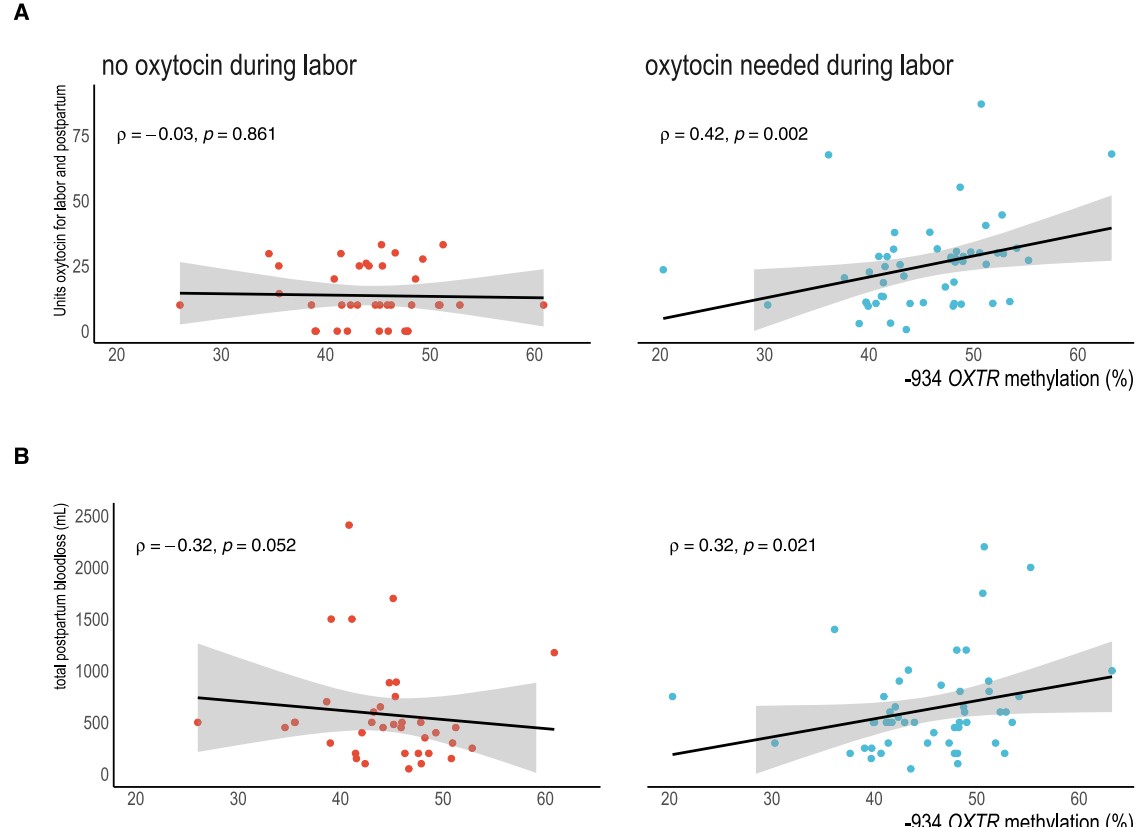

**Fig. 4 Increased *OXTR* DNA methylation is associated with increased oxytocin use and total postpartum blood loss only when oxytocin is administered during labor.** Red dots show participants without oxytocin use in labor ($n = 38$); blue dots denote oxytocin was used in labor ($n = 53$). Shaded area denotes 95% CI of regression line. **A** Higher −934 *OXTR*m is associated with higher intra/postpartum administration of oxytocin among participants receiving oxytocin (blue dots). **B** Postpartum bleeding volume associated with DNA methylation however, severity increased when oxytocin was used in labor.

**Table 4 Increased severity of blood loss and greater relative risk for postpartum hemorrhage were greater when oxytocin was used in labor with higher *OXTR* methylation.**

|  | Study Case (vs. Control) IRR(95%CI) | Cumulative Volume Blood Loss (mL) β(95%CI) | Postpartum Hemorrhage (500 mL) IRR(95%CI) | Postpartum Hemorrhage (1000 mL) IRR(95%CI) |
|---|---|---|---|---|
| Main effects *OXTR*m level (5% increases) without oxytocin in labor | 0.74 (0.59-0.94)† | −109.71 (−207.5 to −11.95)* | 0.71 (0.56-0.90)† | 0.56 (0.35-0.91)* |
| Oxytocin used in labor vs. none and low *OXTR*m | 0.04 (0.003-0.39)* | −1567.57 (−2620.14 to −515.01)† | 0.05 (0.006-0.51)* | 0.001 ($2.00^{-7}$ to 0.09)† |
| Interaction *OXTR*m (5%) X oxytocin use | 1.45 (1.11-1.91)† | 185.22 (70.53-299.91)† | 1.45 (1.12-1.89)† | 2.95 (1.53-5.71)‡ |

*$p < 0.05$
†$p < 0.01$
‡$p < 0.001$
Models included replicate variability, parity, antibiotic administration during labor and body mass index at delivery and used Poisson regression or generalized linear modeling (gamma distribution) for total blood volume.

associated with a RR of 2.95 for postpartum hemorrhage (1000 mL or greater), 95% CI 1.53–5.71. All models controlled for replicate variability, parity, antibiotic use during labor (a proxy for possible intrapartum infection) and BMI at delivery (Table 4).

## Discussion
In this study, we assessed the relationship of *OXTR*m in pregnant uterine myometrium obtained from maternal blood and in a

separate experiment tested the association of *OXTR*m with an individual's need for greater uterine stimulation using oxytocin and subsequent postpartum hemorrhage. The first key finding is that pregnant uterine and blood *OXTR*m are significantly correlated at several sites, providing support for the use of blood *OXTR*m as a surrogate for uterine *OXTR*m. Second, we found a significant positive association between *OXTR*m and greater cumulative oxytocin utilization during parturition. Last, we provide evidence that the severity of postpartum blood loss was

associated with higher *OXTR*m and moderated by oxytocin administration. To the best of our knowledge, this is the first study to examine these associations, which has relevance to understanding an important source of maternal morbidity.

Despite recognized tissue-specific DNA methylation patterns[41], our finding that *OXTR*m derived from uterine myometrium and maternal blood are correlated is encouraging for future use of peripheral blood DNA samples in the ongoing investigation of uterine function during childbirth. Clearly, sampling uterine tissues during childbirth can only be done routinely during surgical (Cesarean) birth, which will necessarily limit the interpretation of the findings to vaginal births. Furthermore, future clinical utility of this kind of biomarker would require it be accessible prior to birth. Future work may determine if salivary *OXTR*m will correlate to uterine levels, as it does with blood[61,62], which would allow for non-invasive measurement of this marker in uterine function. In addition, future study of the variation in *OXTR* expression as a result of *OXTR*m is needed to connect methylation to the functional changes in uterine tissue.

While our work focused only on *OXTR*m, the potential consequences of higher myocyte *OXTR*m include decreased *OXTR* transcription, reduced OXTR density or less effective intracellular signaling (leading to less efficient uterine contraction) leading to diminished pharmacologic responses. In other human studies, differential promoter *OXTR*m has been documented in samples of maternal decidua (higher with preterm labor)[50], in fetal membranes (higher with preterm birth)[58] and in myometrium at term without labor (lower methylation than preterm)[2]. One study in mice showed an association in uterine tissue of higher *Oxtr* promoter methylation and lower mRNA expression, though we note that the *OXTR* CpG site we focused on (−934) is not conserved in mouse models, which may limit translation to human interpretation[96]. Other studies linking higher promoter *OXTR*m to reduced gene transcription have been performed in brain tissue[42,95]. Future research is necessary to examine in vitro uterine tissue contractility in relationship to *OXTR*m and oxytocin administration. In concordance with our findings, other studies have shown a relationship between exogenous oxytocin and diminished physiologic responses with higher *OXTR*m. Among female participants given intranasal oxytocin, Chen *et al.* found fMRI-measured responses in visual cortex were higher in individuals with lower promoter *OXTR*m assessed from saliva[97]. Furthermore, greater promoter *OXTR*m derived from placental vasculature was associated with diminished vasoconstriction in response to exogenous oxytocin[98]. In sum, our study links the peripheral *OXTR*m to key obstetric outcomes as a biomarker, yet requires further investigation in terms of functional differences at the tissue level.

Prior to the hospital admission, obstetric care providers have no indication of how effectively oxytocin will stimulate uterine contractions. Current practice includes titration of the medication by 1–2 mU/min (4–6 mU/min if using a high dose approach) until uterine activity appears regular and the cervix dilates[66]. For some, this may take only a few hours, while others may require many hours or days. Mousa et al. (2008) reported in a sample of over 20,000 births that among individuals who developed postpartum hemorrhage, 10% of women having Cesarean birth and 21% of those with vaginal births failed to respond to uterotonic medication[99]. Whether epigenetic variability of *OXTR* might be responsible for the inability to respond to first line treatments due to lower receptor density has clear clinical relevance, given the variability in clinical oxytocin response both during and after labor. It is also well-established that higher oxytocin use/dosing is needed for people with higher BMI[100], however, whether *OXTR*m plays a role between oxytocin needs and body mass is unknown. Our analyses did account for BMI at the time of birth, which

suggest that *OXTR*m may be a useful predictor of oxytocin needs regardless of body characteristics. More research into DNA methylation (oxytocin sensitivity) could help explain current clinical challenges with labor inductions[101] or lead to developing new personalized strategies for labor management or postpartum hemorrhage prevention/treatment. Possible clinical improvements might include using personalized dosing regimens during labor or alternative uterotonics as first line postpartum hemorrhage prophylaxis. More work will be needed to prospectively test *OXTR*m measured before labor starts for the utility of this biomarker in predicting oxytocin responses both during labor and after giving birth.

**Future research should address *OXTR*m stability over time**. An important consideration when interpreting these findings is whether *OXTR*m is stable throughout a short time frame in parturient females (i.e., birth to 6 weeks postpartum). Given that our sampling was done after birth, we cannot claim that higher *OXTR*m was the mechanism responsible for greater oxytocin needs at parturition. The alternative testable hypothesis is that greater oxytocin usage led to greater *OXTR*m postpartum, an effect magnified by postpartum hemorrhage or higher blood loss, because far more oxytocin would be needed in this scenario. Evidence for this line of reasoning among adult humans is limited; however, oxytocin exposure in the prairie vole has been demonstrated to cause conserved promoter *OXTR*m and gene transcription differences[102]. Assessing *OXTR*m pre-labor as well as postpartum would aide in answering this question. If *OXTR*m was altered by the events and exposures occurring during labor and birth, future questions would address the impact of this change on other oxytocin-dependent outcomes in the transition to mother/parenthood[103].

The sample size for this study was limited by the extenuating circumstances of the pandemic. Study staff also noted that some individuals approached about the study who experienced a much more severe postpartum hemorrhage (> 2000 mL) declined participation due to feeling traumatized by the experience and not wanting to engage with additional activities. Further work on postpartum hemorrhage should consider optimal times to engage individuals in research, given the sensitive nature of this complication. Future work should include gene transcription quantification, which was not possible due to the lack of RNA stabilizing medium at the time of Cesarean birth and institutional research operation limitations. In a larger random sample, future investigations should also consider DNAm variability across maternal age as well as differences by parity as maternal outcomes are strongly linked to these variables. Our sample was also fairly homogenous in terms of maternal race; in more diverse samples, we can examine differences in both DNAm and birth care practices by racial/ethnic demographics.

Primary strengths of this study are the definition of postpartum hemorrhage cases and the sample selection process. Cases were identified not from total blood loss alone (which is known to be typically underestimated)[94] but from several variables indicating a person was bleeding heavily after birth. Furthermore, by design, the sample is comprised of individuals who experienced heavy bleeding most likely due to ineffective uterine contractility. Most large cohort or retrospective studies will use medical record diagnoses of postpartum hemorrhage or billing codes based on the estimated blood loss alone or discharge data, which may be very inaccurate[104]. This approach will group together multiple causes of postpartum hemorrhage into a single outcome, which limits specificity of the associations to oxytocin use or other risk factors. Another strength is the evidence showing a correlation between blood and myometrial *OXTR*m levels. The finding helps

address concerns that blood may not be a viable tissue for understanding the methylation patterns of the uterine tissue. Finally, the pyrosequencing was performed in triplicate and multivariable analyses accounted for variability between replicates. These procedures help ensure that measured *OXTR*m was as reliable and valid as possible.

We present evidence that higher *OXTR*m is associated with greater total oxytocin requirements for labor and postpartum management as well as a significant interaction of *OXTR*m and oxytocin use on higher blood loss due to uterine atony. Together with the findings that maternal blood *OXTR*m is well correlated to uterine *OXTR*m, this study presents interesting findings that contribute to our understanding of variability in maternal health outcomes during parturition.

## Data availability

Source data for all figures are available as a single Excel file, Supplementary Data 1. Data is available upon reasonable request to corresponding author for extensions of this study; however, limited to the participants' data who consented to use in future research during the informed consent process.

## Code availability

No custom code was used to generate or process the data described in the manuscript.

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

## Acknowledgements

We are grateful for the contributions of the participants to this research following a potentially life-threatening birth complication and during the sensitive postpartum period. We also acknowledge research support received from Brenna Park-Egan MPH, Kayla Tabari-House, the Oregon Health and Science University Women's Health Research Unit and Pregnancy Research Team, Placenta Repository Project, especially Kierstyn Tuel, Dr. Leonardo Pereira and Dr. Monica Rincon as well as Dr. Natasha Singh and Professor Mark Johnson (Imperial College, University of London). We are grateful for the mentorship and peer support of the Oregon Building Interdisciplinary Research Careers in Women's Health fellowship throughout this investigation and the ongoing mentorship of Dr. C. Sue Carter. Dr. Erickson was supported during the course of this work in part, by National Institutes of Health grants K12HD043488, K99NR019596-01, CTRC UL1TR002369 as well as a Sigma Theta Tau International Beta-Psi Naomi Ballard Research Award. Dr. Connelly and Dr. Krol's work was supported in part by National Institutes of Health grant R01HD098117. The content is solely the responsibility of the authors and does not necessarily represent the official views of the National Institutes of Health.

## Author contributions

Conceptualization: E.N.E., L.M., J.J.C. Methodology: E.N.E., L.M., J.J.C., K.M.K., J.S.D. Investigation: E.N.E., K.M.K., J.S.D. Visualization: E.N.E., J.S.D. Supervision: E.N.E., L.M., J.J.C. Writing—original draft: E.N.E., K.M.K. Writing—review & editing: E.N.E., J.J.C., L.M., K.M.K., J.S.D.

## Competing interests

The authors declare no competing interests.
