## [Peer Review File · Communications Medicine]

Reviewers' comments:

Reviewer #1 (Remarks to the Author):

Erickson et al. presents an interesting and novel analysis regarding oxytocin methylation amount in maternal blood and postpartum hemorrhage. This seems like a logical extension of their already existing work and an important and timely question regarding biomarkers to predict risk of postpartum hemorrhage. The degree to which a woman in labor responds to oxytocin for augmentation or induction is not predictable nor is their likelihood for postpartum hemorrhage, especially if they have few clinical risk factors.

There are a few suggestions that can enhance this manuscript:

1) Overall: The flow of the introduction into results and materials and methods near the end is a bit hard to follow - especially with some of the methods included in the results (for example lines 170-181)

2) Lines 65-66 - what sort of 'implications' is being referenced here - please clarify.

3) Lines 105-106 - the 2014 definition was changed from 500mL for vaginal delivery and 1000 mL for cesarean delivery to 1000 mL for either vaginal delivery or cesarean delivery - this is not entirely clear here.

4) Line 135 seems a bit too late in the introduction to start with objectives/aims. I think if the introduction were more concise that would help set up the study questions more clearly.

5) Lines 180-181 please clarify how was blood loss measured - clinical estimated blood loss, quantified blood loss or mixed methods?

6) Lines 191-193 please specify what % was oxytocin versus no oxytocin use in cases /controls.

7) Lines 204-204 where it specifies time of blood draw seems too late into the manuscript since I was wondering this the whole time while reading - perhaps if the methods were moved up then this would not be such an issue.

8) Line 233 was PPH > or = 1000mL?

The idea for saliva tests for oxytocin methylation is quite fascinating. I also wonder though if the authors want to speculate how this might affect clinical care if we had developed a biomarker predicting the need for more oxytocin likely in labor or higher risk for PPH - would we consider these patients needing tranexamic acid and/or combined uterotonic prophylactically at delivery for PPH prevention? Or perhaps to start with high dose pitocin protocol so that overall labor is shortened if it is known that the patient likely needs higher doses up front.

Reviewer #2 (Remarks to the Author):

The current manuscript presents novel data regarding the association between administered OT during labor, OXTRm in maternal blood, and postpartum hemorrhage. Additionally, the authors provide important evidence that OXTRm levels in blood and uterine myometrium are correlated, suggesting that blood may serve as a useful proxy-measure of uterine DNA methylation levels. I found the study very interesting and the manuscript was well-written. Below I list several questions and issues that arose while reading the manuscript, which I believe require additional information and clarification, in no particular order of importance:

1. The authors note that study recruitment was halted prematurely due to the Covid-19 pandemic, which lead to a limited sample size. Did the authors conduct a-priori power analyses? It would be helpful if the authors provide information on the initially intended sample size for the study.
2. How are the authors accounting for multiple testing?
3. p. 6 line 163 “Mean OXTRm significantly differed between myometrium and blood (paired t-test), such that myometrial OXTRm levels were lower ($t(25)=10.09$, $p < 0.001$).” Can the authors clarify whether these tests were also only conducted at the -934 locus? Please also specify this in the figure descriptions.
4. The authors briefly mention on p. 6 line 164 that methylation data from the -934 locus are used for the analyses as part of Aim 2. Can the authors provide further justification for focusing on this specific site (e.g. it would be helpful to note that this site is often targeted in OXTRm studies). Please also further delineate that main results are based on methylation at this site.
5. p.7 line 208: “OXTRm was significantly lower in cases compared to controls specifically when labor occurred physiologically” In Table 2 I see that this finding is based on a subsample of $n = 38$, please specify how many cases vs. controls are included in this subsample.
6. p. 8 line 217 and p. 18 line 496: Please elaborate on your choice to use GLMs with a gamma distribution after using linear regressions
7. p. 8 line 217: I understand from Table 3 that the GLM was conducted among the subsample of participants that received OT during labor. It would be helpful if this is also clarified in the text.
8. P. 8 line 227-228: “When oxytocin was used during labor” – when I first read this it seemed that analysis were conducted in a subsample of participants that received oxytocin during labor (similar to the correlations reported at the beginning of this section). However, from Table 4 it becomes clear that these analyses were conducted in all participants ($n=91$). It would be helpful to clarify this in the text.

9. Please check p. 8 line 232-233: “the combination of higher OXTRm and oxytocin use in labor was associated with a RR of 2.95 for PPH (1000mL), 95% CI 1.53-5.71.” In table 4 the statistics are stated as 1.95 (1.53-5.71)

10. The authors provide an important consideration in the discussion about the stability of OXTR methylation across time. In the current study, the relationship between OXTRm in uterine tissue and blood was detected in blood drawn just prior to labor. Since the authors suggest that OT usage may lead to changes in methylation post-partum, I believe an important addition here may also be that it remains uncertain if OXTRm in maternal blood drawn 6-10 weeks after birth—as is done for Aim 2—relates to OXTRm levels in uterine tissue. Additionally—as the authors also mention on p.10 line 291-292—since sampling was done after birth, this means that OXTRm levels cannot be claimed as responsible for factors at parturition. As such, the authors should be careful with sentences that suggest a causal direction, i.e. in the abstract where it is stated that OXTRm “is linked to pharmacologic oxytocin requirements during parturition, which together influences subsequent postpartum hemorrhage.”

11. The authors did not take maternal smoking into consideration. I believe this limitation should be addressed, since smoking can lead to extensive changes in DNA methylation and is associated with negative pregnancy outcomes.

Reviewer #3 (Remarks to the Author):

The manuscript of Erickson et al. describes an interesting study assessing the correlation between OXTR methylation and postpartum blood loss. The underlying hypothesis is that given the role of OT in parturition, reduced sensitivity to OT (mediated via methylation control of OXTR gene expression), predisposes to PPH. Further, because peripheral OXTR methylation matches uterine methylation patterns the former can be used as a proxy measure in the form of a diagnostic test of a blood sample. This is a coherent and interesting idea that one could see would be of use when planning inductions of labour. Overall, the manuscript is well written, concise and contains an appropriate level of supportive and critical discussion of results. The results themselves are well presented and appropriately analysed. I have no real concerns about the data as presented. However, given the potential benefit of the study I do feel that a further set of experiments would significantly enhance the confidence one could have in the general conclusion. The issue at hand is well identified in the sentence beginning end of 287 "When oxytocin was used during labor, each 5% higher OXTRm level was associated with 45% higher relative risk of being a case compared to a control (IRR 1.45, 95% CI = 1.11-1.91)." The question is what does a 5% higher OXTRm mean in terms of gene expression? In the first validating part of the study the authors had matched myometrial and peripheral blood samples, in which, they measured OXTRm status. Why did they not also measure OXTR mRNA? This is relatively trivial experiment to do and yet would have yielded invaluable information. One could construct an OXTRm vs [OXTR] mRNA relationship and then be able to say something meaningful about what a 5% higher OXTRm means. If a 5% change in OXTRm is a significant change in OXTR mRNA level, and the relationship between OXTRm and [OXTR] mRNA is a tight correlation, then the authors primary hypothesis would be strongly supported.

Reviewer #4 (Remarks to the Author):

The manuscript entitled “Epigenetic state of oxytocin receptor is associated with exogenous oxytocin needs during human birth and increased risk of postpartum hemorrhage” attempted to draw a correlation between levels of DNA methylation at the OXTR gene and postpartum hemorrhage. This is a very important topic given that postpartum hemorrhage is the leading cause of death associated with pregnancy. The identification of prognostic markers that would enable patients to be identified prior to labor for their risk of hemorrhage would greatly improve clinical intervention and patient survival. However, on its current form, the manuscript does not provide strong evidence of such correlation.

From a general perspective, the manuscript is poorly organized and presents the structure of a structure of a thesis-like report, rather than a document that is ready to be published. The introduction is long and repetitive. Several parts of the result session should be transferred to material and methods (patient info/enrollment, sample collection, etc), and replaced by better and appropriated data interpretation.

But most importantly, the analysis discussed in this manuscript are superficial. For example, none of the analysis show the levels of DNA methylation at the OXTR across samples, CpG islands, or control versus patient. There is not even a genome browser image properly showing levels of DNA methylation. Plus, a 30% to 40% change on the levels of DNA methylation may not be meaningful in terms of gene expression regulation – which was not tested. These issues collectively make impossible for this review to fully appreciate the analysis, and to correlate it with the author’s conclusions. The authors made no attempt to link levels of DNA methylation with OXTR mRNA levels or with any other parameter that could also be involved with hemorrhage (coagulation factors, platelets level, etc). Accordingly, there was no hematological analysis to define blood cells changes in response to OTX administration, which could also contribute to hemorrhage control. These are fundamental points that need to be evaluated before any conclusion can be drawn from the current analysis.

Suggestions for Edits and clarity:

Page 1:

Line 2-3: Title is unclear whether state of receptor and “needs” are in infant or mother

Line 22-23: “Evaluate blood as a surrogate for uterine” is confusing, maybe as “an indicator of... [uterine methylation status.]”

Line 29: “in the OXTR is linked..”

Line 30: “which together influences subsequent postpartum hemorrhage.” This conclusion is not very clear nor strong.

Lines 35-37: The teaser needs work rewording, as it does not matches the conclusions from the abstract or the article. No clear cause/effect is established in the paper, so sensitivity to oxytocin doesn’t seem like the correct verbiage.

Page 2:

Line 60: add comma after “head/body, triggering..”

Line 76: specify what autism/social cognition is in reference to fetal/infant health, as other conditions reported impact maternal health

Page 3:

Line 80: it is unclear whether the methylated DNA under consideration is maternal or fetal/infant, given that fetal blood cells can be detected in the mother's circulation.

Line 97: Need citation to support the claim that the rates of labor induction are on the rise.

Line 97: remove semi-colon after “...oxytocin administration; most...” and replace with comma.

Lines 99 and 103: Clarification: oxytocin is a first-line therapy for hemorrhage prevention and treatment, but also increases risk of postpartum hemorrhage?

Page 4:

Line 108: remove comma after “...life-saving interventions, (blood transfusion....),” as there is already one after the close of the parentheses.

Line 122: comma after “...prolonged labor, or...”

Page 5:

Line 144: Provide insight into hypothesis? What were findings?

Lines 144-146: This last sentence is fairly vague in terms of study importance in a clinical setting or from maternal/infant health perspective

Line 152-155: Is this mentioned in discussion the limitations of using a multiparous aged cohort? How is hemorrhage and/or need for oxytocin, or level of methylation, correlated to these characteristics?

Page 6:

Lines 162-166: Rationale nor data to support the selection of site -934 for further analysis is not completely clear..

Line 188: So cases of pp hemorrhage had higher BMI, and higher needs for oxytocin. How does obesity impact oxytocin production, receptor expression, and methylation? Relates to the conclusion on Line 193 that quantity, duration, and max dosage was higher. The authors show no correlation at delivery between BMI and OXTR methylation (Line 210, page 7), but doesn't necessarily mean there's no correlation with oxytocin itself, or receptor status?

Page 7:

Lines 193-204: conclusions seem to be very intuitive based on the general definition of “hemorrhage” compared to non-hemorrhagic bleeding... ie, amount of blood and the need for transfusion. Not novel.

Page 11:

Line 301: please remove the reference to the pandemic throughout the paper. The limitations section does not seem to really address tangible, scientific limitations of the study (ie, identity of cohort). This is ok on a thesis, not on a manuscript

Figures and Tables:

Figure 2: How do the individual points match up between individual patients with respect to the two samples?

Table 1: Interesting point on the racial disparity in hemorrhaging patients which were "Latin American" or "African", but not fully developed or discussed.

Figure 3: It is unclear whether this correlation is actually strong based on data. Also, is the conclusion that more oxytocin given increases methylation, or rather than increased methylation means more oxytocin is required. Wouldn't the expectation be that increased oxytocin activates receptor expression and therefore loss of DNA methylation?

Overall, conclusions are muddled and not clearly stated, especially in terms of how findings are relevant to clinical applications.

Specific Outstanding questions and issues:

How does oxtr methylation actually impact oxytocin signaling? Physiologically how is this relevant to the need for increased oxytocin administration, etc.

Show higher blood loss but how does this actually relate to things like, risk of future hemorrhage during future pregnancy, or fatality, etc. state implications in understanding maternal morbidity, but do not discuss morbidity or fatality at all (line 245, page 9)

Discussion of uterine function and contractility, but not of how oxytm and hemorrhaging are related. Connection with hemorrhaging not fully convincing.

Other cohort issues: Race disparity, BMI differences, gestational diabetes rate, infection and antibiotics. Unclear how these factors were controlled for (stated in line 234).

Cohort from aim 1 is mainly multiparous, whereas aim 2 is largely nulliparous. How might this discrepancy impact this study?

General mentions of uterine contractility and how this relates to oxytocin and hemorrhaging could use a bit more development.

Reviewers' comments:

Reviewer #1 (Remarks to the Author):

Erickson et al. presents an interesting and novel analysis regarding oxytocin methylation amount in maternal blood and postpartum hemorrhage. This seems like a logical extension of their already existing work and an important and timely question regarding biomarkers to predict risk of postpartum hemorrhage. The degree to which a woman in labor responds to oxytocin for augmentation or induction is not predictable nor is their likelihood for postpartum hemorrhage, especially if they have few clinical risk factors.

There are a few suggestions that can enhance this manuscript:

1. Overall: The flow of the introduction into results and materials and methods near the end is a bit hard to follow - especially with some of the methods included in the results (for example lines 170-181)
 - a. **Thank you, we have updated the formatting for this paper, the order being: introduction, methods, results. We have separated more clearly the information noted in lines 170-181 into areas in the methods where appropriate.**
2. Lines 65-66 - what sort of 'implications' is being referenced here - please clarify.
 - a. **Thank you: This sentence was broken into two: The regulation of OXTR in the myometrium has been investigated for its role in labor processes⁴ with the quantity and responsiveness of OXTR playing an important role in uterine contraction during labor.²¹ Furthermore, variability in OXTR function could also influence the postpartum outcomes of uterine atony and subsequent postpartum hemorrhage.²² (lines 73-77).**
3. Lines 105-106 - the 2014 definition was changed from 500mL for vaginal delivery and 1000 mL for cesarean delivery to 1000 mL for either vaginal delivery or cesarean delivery - this is not entirely clear here.
 - a. **Thank you for pointing this out- we have updated this for clarity (lines 78-79).**
4. Line 135 seems a bit too late in the introduction to start with objectives/aims. I think if the introduction were more concise that would help set up the study questions more clearly.
 - a. **Thank you for this feedback: given the broader nature of the audience, we have been more descriptive in the introduction in general, than if the audience were a purely obstetric audience or those with expertise on DNA methylation for example. We have edited throughout the introduction to reduce non-vital background information as best as possible and reorganized it substantially to bring together the concepts more efficiently.**
5. Lines 180-181 please clarify how was blood loss measured - clinical estimated blood loss, quantified blood loss or mixed methods?
 - a. **We added a line in the methods to help explain that there are mixed methods of measurement in the study, given the variety of settings and providers as well as standards by institutions (line 220-25).**
6. Lines 191-193 please specify what % was oxytocin versus no oxytocin use in cases /controls.
 - a. **We added this information into the results section: "did not differ between cases (n = 42, 60.9%) and controls (n = 28, 56.0%)" (line 415-16).**

7. Lines 204-204 where it specifies time of blood draw seems too late into the manuscript since I was wondering this the whole time while reading - perhaps if the methods were moved up then this would not be such an issue.
 - a. **We agree, the methods have been moved up.**
8. Line 233 was PPH > or = 1000mL?
 - a. **We have updated this line to clarify the findings (now line 454).**
9. The idea for saliva tests for oxytocin methylation is quite fascinating. I also wonder though if the authors want to speculate how this might affect clinical care if we had developed a biomarker predicting the need for more oxytocin likely in labor or higher risk for PPH - would we consider these patients needing tranexamic acid and/or combined uterotonics prophylactically at delivery for PPH prevention? Or perhaps to start with high dose pitocin protocol so that overall labor is shortened if it is known that the patient likely needs higher doses up front.
 - a. **Thank you for seeing the potential for clinical improvement from this work, we have similar speculative ideas as well. We added a line to reflect some of the more specific long-term implications of this line of study. "Possible clinical improvements might include using personalized dosing regimens during labor or alternative uterotonics as first line PPH prophylaxis." (line 513).**

Reviewer #2 (Remarks to the Author):

The current manuscript presents novel data regarding the association between administered OT during labor, *OXTRm* in maternal blood, and postpartum hemorrhage. Additionally, the authors provide important evidence that *OXTRm* levels in blood and uterine myometrium are correlated, suggesting that blood may serve as a useful proxy-measure of uterine DNA methylation levels. I found the study very interesting and the manuscript was well-written. Below I list several questions and issues that arose while reading the manuscript, which I believe require additional information and clarification, in no particular order of importance.

1. The authors note that study recruitment was halted prematurely due to the Covid-19 pandemic, which lead to a limited sample size. Did the authors conduct a-priori power analyses? It would be helpful if the authors provide information on the initially intended sample size for the study.
 - a. **Yes, we had performed this a-priori and added this into the text/methods section: "Recruitment goal was 100 cases and 100 matched controls based on a sample size calculation of a mean difference between groups of a magnitude of two percentage points (SD of 5.0) with 80% power and α of 0.05. Of the 577 postpartum records screened, we approached 393 individuals and 120 met criteria and consented to participate in the study." (line 202).**
2. How are the authors accounting for multiple testing?
 - a. **We applied a Benjamini Hochberg corrected to the multiple testing within the CpG sites we analyzed for correlation between tissues. We arrived at the same set of significant CpGs, though the p-values were modified. (line 336).**
3. p. 6 line 163 "Mean *OXTRm* significantly differed between myometrium and blood (paired t-test), such that myometrial *OXTRm* levels were lower ($t(25)=10.09$, $p < 0.001$)." Can the authors clarify whether these tests were also only conducted at the -934 locus? Please also specify this in the figure descriptions.

- a. **Originally this was the only comparison between the tissues, however, we added comparisons between all the significant sites for consistency and adjusted for multiple testing and denoted this comparison using a new Figure 2. A Bonferroni correction was applied to the comparison of DNA methylation between myometrium/ blood samples (Figure 2). We removed the mean methylation (by tissue) comparisons from the text, as it was not the primary hypothesis of the experiment however.**
4. The authors briefly mention on p. 6 line 164 that methylation data from the -934 locus are used for the analyses as part of Aim 2. Can the authors provide further justification for focusing on this specific site (e.g. it would be helpful to note that this site is often targeted in OXTRm studies). Please also further delineate that main results are based on methylation at this site.
 - a. **Thank you, we added the following detail: for the case-control study, given that the site -934 (chr3: 8,810,729-8,810,845) demonstrated both high statistical significance between tissues, is linked to differences in transcription of *OXTR*, and is a commonly studied CpG site in human adult literature, we chose to focus on this site for our examination of oxytocin administration and postpartum hemorrhage outcomes. (Lines starting 386)**
5. p.7 line 208: “OXTRm was significantly lower in cases compared to controls specifically when labor occurred physiologically” In Table 2 I see that this finding is based on a subsample of n = 38, please specify how many cases vs. controls are included in this subsample.
 - a. **Yes, we added this detail “Using a linear regression model and controlling for pyrosequencing replicate variability, we found *OXTRm* to be significantly lower in cases compared to controls (n = 22 cases and 16 controls) specifically when labor occurred physiologically (oxytocin was not used) (4.15% lower; 95% CI -7.91- -0.39) (Table 2).” Line 420**
6. p. 8 line 217 and p. 18 line 496: Please elaborate on your choice to use GLMs with a gamma distribution after using linear regressions.
 - a. **Thank you- we updated the text for more description: “As the distribution of oxytocin use in labor was significantly right-skewed (Shapiro-Wilk *W* test for normality $W=0.87$, $p<.0001$), we used a multivariable generalized linear model (GLM) with a gamma distribution to examine oxytocin use by *OXTRm* in lieu of a linear regression or transforming the data..” (line 431)**
7. p. 8 line 217: I understand from Table 3 that the GLM was conducted among the subsample of participants that received OT during labor. It would be helpful if this is also clarified in the text.
 - a. **Yes, this is clarified in the text as above. (line 431 on)**
8. P. 8 line 227-228: “When oxytocin was used during labor” – when I first read this it seemed that analysis were conducted in a subsample of participants that received oxytocin during labor (similar to the correlations reported at the beginning of this section). However, from Table 4 it becomes clear that these analyses were conducted in all participants (n=91). It would be helpful to clarify this in the text.
 - a. **Thank you we clarified this more explicitly in this section of the text. “Given that oxytocin use appeared to be an effect modifier, we used Poisson regression and GLM models with interactions to examine this more fully across the entire sample. “(Line 445)**

9. Please check p. 8 line 232-233: “the combination of higher OXTRm and oxytocin use in labor was associated with a RR of 2.95 for PPH (1000mL), 95% CI 1.53-5.71.” In table 4 the statistics are stated as 1.95 (1.53-5.71)
- a. **Yes, thank you for catching the typo, it is RR 2.95, we ran the model again to be certain and updated the table. The text was correct (line 454 now).**
10. The authors provide an important consideration in the discussion about the stability of OXTR methylation across time. In the current study, the relationship between OXTRm in uterine tissue and blood was detected in blood drawn just prior to labor. Since the authors suggest that OT usage may lead to changes in methylation post-partum, I believe an important addition here may also be that it remains uncertain if OXTRm in maternal blood drawn 6-10 weeks after birth—as is done for Aim 2—relates to OXTRm levels in uterine tissue. Additionally—as the authors also mention on p.10 line 291-292—since sampling was done after birth, this means that OXTRm levels cannot be claimed as responsible for factors at parturition. As such, the authors should be careful with sentences that suggest a causal direction, i.e. in the abstract where it is stated that OXTRm “is linked to pharmacologic oxytocin requirements during parturition, which together influences subsequent postpartum hemorrhage.”
- a. **Thanks for this observation. Yes, this is an important question that will require further study. We do state in the paragraph referenced by the reviewer that we cannot claim the OXTR DNAm was responsible for the outcome. The phrasing listed above was intended to reference the interaction tests performed (i.e. the word “together”) however, we have modified this text in the abstract to avoid any misleading or overstated descriptions of our findings. “We provide the first evidence that epigenetic variability in OXTR is associated with pharmacologic oxytocin requirements during parturition and moderates subsequent postpartum hemorrhage” (line 30)**
11. The authors did not take maternal smoking into consideration. I believe this limitation should be addressed, since smoking can lead to extensive changes in DNA methylation and is associated with negative pregnancy outcomes.
- a. **We added the smoking history/present smoking variable to the table for background detail. It was recorded by self-report and also in the medical record abstraction. Smoking did not differ between cases/controls (table 1), nor did the mean DNA methylation differ between cases/controls(not reported in paper). Smoking status was not available for the myometrial sample participants.**

Reviewer #3 (Remarks to the Author):

1. The manuscript of Erickson et al. describes an interesting study assessing the correlation between OXTR methylation and postpartum blood loss. The underlying hypothesis is that given the role of OT in parturition, reduced sensitivity to OT (mediated via methylation control of OXTR gene expression), predisposes to PPH. Further, because peripheral OXTR methylation matches uterine methylation patterns the former can be used as a proxy measure in the form of a diagnostic test of a blood sample. This is a coherent and interesting idea that one could see would be of use when planning inductions of labour. Overall, the manuscript is well written, concise and contains an appropriate level of supportive and critical discussion of results. The results themselves are well presented and appropriately analysed. I have no real concerns about the data as presented.
- a. **Thank you for this encouraging feedback.**

2. However, given the potential benefit of the study I do feel that a further set of experiments would significantly enhance the confidence one could have in the general conclusion. The issue at hand is well identified in the sentence beginning end of 287 "When oxytocin was used during labor, each 5% higher OXTRm level was associated with 45% higher relative risk of being a case compared to a control (IRR 1.45, 95% CI = 1.11-1.91)." The question is what does a 5% higher OXTRm mean in terms of gene expression? In the first validating part of the study the authors had matched myometrial and peripheral blood samples, in which, they measured OXTRm status. Why did they not also measure OXTR mRNA? This is relatively trivial experiment to do and yet would have yielded invaluable information. One could construct an OXTRm vs [OXTR] mRNA relationship and then be able to say something meaningful about what a 5% higher OXTRm means. If a 5% change in OXTRm is a significant change in OXTR mRNA level, and the relationship between OXTRm and [OXTR] mRNA is a tight correlation, then the authors primary hypothesis would be strongly supported.
- a. **We agree with this reviewer's comments and state this as a limitation of the findings in the discussion. The simple answer is that the myometrial samples were not preserved in RNA stabilizing medium (they were processed at room temperature for various lengths of time (up to an hour) prior to being frozen in liquid nitrogen) and it was felt that results may be unreliable or uninterpretable from this tissue because of poor RNA quality. That being said, our team has previously published evidence in both humans and prairie voles that blood-derived OXTRm at CpG -934 is negatively associated with OXTR expression.**
 - b. **However, due to reviewers' concerns did take time to attempt to quantify OXTR expression. However, unfortunately, we were able to isolate RNA from 16 of the myometrial samples with RNA Integrity Numbers (RIN) ranging from 3.2-8.5.**
 - c. **The overall correlation between level of methylation and OXTR expression was non-significant (R = 0.10, p = 0.70), we also adjusted for the RNA quality using the RIN in a regression model with OXTR expression as the DV, again with a NS relationship between -934 methylation and OXTR expression.**
 - d. **We next conducted an exploratory analysis on the interaction between a commonly studied OXTR SNP (rs53576) and methylation and did find that the direction of the relationship between OXTR expression and methylation was associated with genotype. In effect, the rs53576 A-carrier participants (n=6) had lower OXTR expression when OXTR methylation was higher (B= -85.2, 95% CI -161.7 - -8.6) relative to G/G (n =10) participants, adjusting for the RIN value. We also explored how blood methylation might relate to OXTR expression as well—finding similar significant findings for the interaction effect (A-carriers), but not across the whole set of 16 participants with RNA.**
 - e. **We believe this interesting finding warrants confirmation in future work, however, is outside the scope of this paper given missing RNA data and potential concerns over RNA quality.**
 - f. **In sum, the mechanistic experiments linking OXTR methylation to expression or other molecular/functional receptor changes is the goal of our future work, as we cannot fully address this limitation with these data.**

Reviewer #4 (Remarks to the Author):

The manuscript entitled “Epigenetic state of oxytocin receptor is associated with exogenous oxytocin needs during human birth and increased risk of postpartum hemorrhage” attempted to draw a correlation between levels of DNA methylation at the OXTR gene and postpartum hemorrhage. This is a very important topic given that postpartum hemorrhage is the leading cause of death associated with pregnancy. The identification of prognostic markers that would enable patients to be identified prior to labor for their risk of hemorrhage would greatly improve clinical intervention and patient survival. However, on its current form, the manuscript does not provide strong evidence of such correlation.

1. From a general perspective, the manuscript is poorly organized and presents the structure of a structure of a thesis-like report, rather than a document that is ready to be published.
 - a. **Thank you for the feedback, we hope revisions in the structure of the paper and additions we have made to the manuscript will be useful in improving the readability of the study for the reviewer.**
2. The introduction is long and repetitive.
 - a. **We substantially revised the introduction to be avoid repetition and to provide a better flow of the ideas.**
3. Several parts of the result session should be transferred to material and methods (patient info/enrollment, sample collection, etc), and replaced by better and appropriated data interpretation.
 - a. **The manuscript has been updated in regards to formatting, the methods were moved after the introduction and rearranged accordingly, better separating methods from results for clarity.**
4. But most importantly, the analysis discussed in this manuscript are superficial. For example, none of the analysis show the levels of DNA methylation at the OXTR across samples, CpG islands, or control versus patient.
 - a. **We have added bivariate comparisons alongside the multivariate models (lines 415, 420,) in addition to a new Figure 3, which shows the comparison visually.**
5. There is not even a genome browser image properly showing levels of DNA methylation.
 - a. **Given that we are looking at a handful of specific sites that were available through the Illumina 850K platform and also candidate pyrosequencing, we are not clear on how the genome browser image would provide additional utility beyond Figure 1, which shows a schematic of OXTR.**
6. Plus, a 30% to 40% change on the levels of DNA methylation may not be meaningful in terms of gene expression regulation – which was not tested.
 - a. **Addressed gene expression limitations above under reviewer #3 .**
7. These issues collectively make impossible for this review to fully appreciate the analysis, and to correlate it with the author’s conclusions.
 - a. **We believe that the revisions we've made, based on the reviewer's aforementioned suggestions, will help readers understand our analyses and conclusions**
8. The authors made no attempt to link levels of DNA methylation with OXTR mRNA levels or with any other parameter that could also be involved with hemorrhage (coagulation factors, platelets level, etc).

- a. We have indicated that the lack of mRNA as a known limitation in the manuscript (lines 530), please see further detail of these experiments in response to reviewer 3. This mechanistic work is the aim of our future studies.
- b. Our reason for examining DNA methylation of *OXTR* was to target our hypothesis on the fact that uterine atony (ineffective uterine contraction) is a primary causal factor in developing postpartum hemorrhage. We would not have expected, nor hypothesized, that DNA methylation of *OXTR* – presumably impacting contractility – would impact coagulation directly. To try to exclude non-uterine atony sources of bleeding, we excluded *a priori* people with known coagulation disorders or significant thrombocytopenia as noted in the methods, we also excluded significant bleeding occurring from genital lacerations, which may have also indicated a bleeding issue. However, due to the case-control nature of study (enrolling postpartum), an extensive pre-birth coagulation panel would have been implausible. The nature of collecting the DNA methylation sample after birth would be difficult to interpret alongside a postpartum platelet count in terms of what had occurred during the birth, as platelets vary significantly from pregnancy to postpartum.
- c. That being said, in response to the reviewer’s comment we looked into this further. We had 6 participants where mild gestational thrombocytopenia was noted, all with platelets >80,000 prior to giving birth. Two were controls (4%) and 4 were cases (5.8%), $p = 0.66$. Four had DNA analyzed. Upon receipt of the reviewer’s comment, we ran an additional regression model on the bleeding outcomes with mild thrombocytopenia as a covariate and found the results were materially the same for the main outcomes listed in Table 4:

	Case (vs. Control) IRR(95%CI)	Total Blood Loss β (95%CI)	Postpartum Hemorrhage (500 mL) IRR(95%CI)	Postpartum Hemorrhage (1000 mL) IRR(95%CI)
Main effects				
OXTRm level (5%)	0.74 (0.59-0.94)†	-112.8 (-210.20- -15.44)*	0.71 (0.56-0.90)†	0.59 (0.36-0.96)*
Oxytocin used in labor vs. none	0.04 (0.003-0.38)*	-1581.40 (-2640.38- -522.42)†	0.05 (0.006-0.50)*	0.0004 (6.28 ⁻⁷ - 0.31)†
Interaction				
OXTRm X oxytocin use	1.46 (1.11-1.91)†	186.49 (70.86-302.11)†	1.45 (1.12-1.89)†	2.56 (1.30-5.04)†

* $p < 0.05$, † $p < 0.01$, ‡ $p < 0.001$

9. Accordingly, there was no hematological analysis to define blood cells changes in response to OTX administration, which could also contribute to hemorrhage control. These are fundamental points that need to be evaluated before any conclusion can be drawn from the current analysis
 - a. **We are not aware of any study that indicates that OXT causes blood cell counts to change fundamentally that would cause (or inhibit) bleeding. The mechanism being tested in this study is the role OXTR plays in uterine contractility as the primary site of action for OXT administration.**
10. Suggestions for Edits and clarity: Page 1:Line 2-3: Title is unclear whether state of receptor and “needs” are in infant or mother

- a. **Thank you for this suggestion—we have changed the term birth to parturition to reflect that the action of oxytocin being studied is within the mother’s body**
11. Line 22-23: “Evaluate blood as a surrogate for uterine” is confusing, maybe as “an indicator of... [uterine methylation status.]”
- a. **Thank you, edit made. (line 23)**
12. Line 29: “in the OXTR is linked..”, Line 30: “which together influences subsequent postpartum hemorrhage.” This conclusion is not very clear nor strong.
- a. **This statement describes the interaction models from the case control study, which indicate that oxytocin administration and OXTRm together result in higher blood loss/PPH. We have edited to help limit confusion. (line 31)**
13. Lines 35-37: The teaser needs work rewording, as it does not matches the conclusions from the abstract or the article. No clear cause/effect is established in the paper, so sensitivity to oxytocin doesn’t seem like the correct verbiage.
- a. **Edited.**
14. Page 2: Line 60: add comma after “head/body, triggering..”
- a. **Done**
15. Line 76: specify what autism/social cognition is in reference to fetal/infant health, as other conditions reported impact maternal health
- a. **We reorganized this for clarity of phrasing (lines 103-4).**
16. Page 3:Line 80: it is unclear whether the methylated DNA under consideration is maternal or fetal/infant, given that fetal blood cells can be detected in the mother's circulation.
- a. **This is a very interesting comment; however, the majority of genomic DNA is maternal (see <https://www.sciencedirect.com/science/article/pii/S0002929707609677?via%3Dihub>) . Fetal DNA (which is found in the form of cell free DNA in maternal blood) represents 6% of DNA in plasma during pregnancy (Lo et al. 1998), which falls to undetectable levels by 2 hours postpartum (mean half-life of 16.3 minutes; Lo et al. 1999). Given that the DNA was sampled at least six weeks postpartum, the quantity in circulation- if present at all- would be negligible. In healthy pregnancy, whole fetal blood cells are not passing through the placental membrane into maternal circulation unless there were significant antenatal hemorrhage at the site of the placenta (trauma/placental abruption, for example, before birth occurred) these would be quite unlikely to be found in postpartum circulation at 6-10 weeks after birth. We did not enroll anyone who had a placental abruption occurring during pregnancy.**
17. Line 97: Need citation to support the claim that the rates of labor induction are on the rise.
- a. **Citation added and data in text. “Oxytocin administration is the primary pharmacological intervention used during the course of labor^{61,62} and is frequently used for labor induction,⁶³ rates of which have risen from 9.5% of births in 1990⁶⁴ to 31.4% of births in 2020.⁶⁵”(lines 119-121)**
18. Line 97: remove semi-colon after “...oxytocin administration; most...” and replace with comma.
- a. **N/A Line has been changed in course of organizing the introduction.**
19. Lines 99 and 103: Clarification: oxytocin is a first-line therapy for hemorrhage prevention and treatment, but also increases risk of postpartum hemorrhage?
- a. **Yes, these statements are accurate—oxytocin when used during labor is associated with increased PPH outcomes, however, oxytocin is the first drug utilized when trying**

to prevent or treat PPH—both time points work on uterine contractility for different purposes. We have rephrased this section for clarity. (Lines 123-136)

20. Page 4: Line 108: remove comma after “...life-saving interventions, (blood transfusion....),” as there is already one after the close of the parentheses.
 - a. **Done (now line 33-34)**
21. Line 122: comma after “...prolonged labor, or...”
 - a. **We can’t find this one, we presume this was from a section that has been deleted or reorganized at present.**
22. Page 5: Line 144: Provide insight into hypothesis? What were findings?
 - a. **Unfortunately, we were unclear as to what this comment is referring to.**
23. Lines 144-146: This last sentence is fairly vague in terms of study importance in a clinical setting or from maternal/infant health perspective
 - a. **Clinical risk assessment is conducted on an ongoing basis throughout a childbirth encounter therefore, is an important part of maternal / obstetric health care and will influence outcomes. We have updated the organization of the introduction to help make this point more clearly. Limitations of current models of risk assessment is detailed starting line 137.**
24. Line 152-155: Is this mentioned in discussion the limitations of using a multiparous aged cohort? How is hemorrhage and/or need for oxytocin, or level of methylation, correlated to these characteristics?
 - a. **We are unclear as to how to address this comment as multiparity and age are not synonymous. The tissue-matching experiment sample did not have sufficient nulliparous donors to allow for methylation comparison by parity. We matched for parity in the case-control study enrollment by design. However, as multiparous individuals were less likely to have hemorrhage in the case-control design in the end, we added parity to the models to further control for any influence of parity on hemorrhage outcome. We added some additional comparisons to Table 2—to demonstrate that we considered other possible associations between the sample’s characteristics and *OXTR* level. As visible in the table, none of the comparisons were significant (age, parity, BMI, gestational age, infant sex or smoking status, anemia). (line 423 and Table 2)**
25. Page 6: Lines 162-166: Rationale nor data to support the selection of site -934 for further analysis is not completely clear..
 - a. **We addressed this comment in response to Reviewer 2 above. Thank you, we added the following detail: In the case/control study, given that the site -934 (chr3: 8,810,729-8,810,845) demonstrated both high statistical significance between tissues, is linked to differences in transcription of *OXTR*, and is a commonly studied CpG site in human adult literature, we chose to focus on this site for our examination of oxytocin administration and postpartum hemorrhage outcomes. (Lines starting 394)**
26. Line 188: So cases of pp hemorrhage had higher BMI, and higher needs for oxytocin. How does obesity impact oxytocin production, receptor expression, and methylation? Relates to the conclusion on Line 193 that quantity, duration, and max dosage was higher. The authors show no correlation at delivery between BMI and *OXTR* methylation (Line 210, page 7), but doesn’t necessarily mean there’s no correlation with oxytocin itself, or receptor status?

- a. **There are clinical studies (cited in our text- studies by Carlson, Maeder, among others) showing higher oxytocin needs among individuals with higher BMI, this is shown in several studies, including our data here. As such, we control for BMI at delivery in the regression analysis to adjust for the association between greater OXT use and higher body mass. We added to Table 2, showing that *OXT* itself was not correlated with BMI. Endogenous oxytocin production and BMI are an interesting consideration, however as we did not sample oxytocin production, we cannot provide further comment, unfortunately.**
27. Page 7: Lines 193-204: conclusions seem to be very intuitive based on the general definition of “hemorrhage” compared to non-hemorrhagic bleeding... ie, amount of blood and the need for transfusion. Not novel.
- a. **Yes, we agree, it shouldn’t be novel. These descriptions are not conclusions, they are describing the features of our cases and controls to demonstrate the groups are distinct in terms of their bleeding and treatment profiles. Determining PPH during or after birth is often a subjective diagnosis, and subject to high variability by care providers, so these comparisons lend weight to our well phenotyped sample by cases and controls. To avoid confusion, chose to streamline this description starting at line 392 and are now referring the reader to the Table for review of the sample in greater detail.**
28. Page 11: Line 301: please remove the reference to the pandemic throughout the paper. The limitations section does not seem to really address tangible, scientific limitations of the study (ie, identity of cohort). This is ok on a thesis, not on a manuscript
- a. **Respectfully, we disagree. Science (and health care) does not occur devoid of influence from the outside world—some of the limitations (like DNA availability) were not due to a lack of effort, lack of interest or technical issues. This information helps put the limitations in context for the reader. If patients were not signing up for the study because of other factors, we would expect to read about it. We also acknowledged, for example, that many people who had very severe PPH (>2000mL) declined participation because of feelings of trauma and not wanting to talk about their experience, which limited our sample and the severity of bleeding in the PPH subgroup. Our target sample was 200, failing to address why this was not reached leaves the reader speculating as to why. Full transparency in this unprecedented situation does not detract from the merits of the findings.**
29. Figures and Tables: Figure 2: How do the individual points matchup between individual patients with respect to the two samples?
- a. **We updated the figures to show the correlation between tissues using a scatter plot, which should help answer this question. (Figure 2)**
30. Table 1: Interesting point on the racial disparity in hemorrhaging patients which were “Latin American” or “African”, but not fully developed or discussed.
- a. **The overall sample was not tremendously diverse, and with only n=1 control being of Latina American ancestry (2%)—while statistically different from cases, we did not feel the limited control group was sufficient to develop a lot of discussion—however, it would be very useful to examine in future, more robust samples with diverse populations.**
31. Figure 3: It is unclear whether this correlation is actually strong based on data.

- a. **Figure 3 has been updated to visualize differences between cases/controls based on oxytocin administration during labor- which supports the use of interaction models. Figure 4 (top) presents the two primary findings the case control study, that Units of oxytocin required was higher based on DNA methylation and (bottom) that the relationship between oxytocin use and blood loss was moderated by DNA methylation levels- the accompanying multivariate models for these graphs are presented in Table 4. The effect of the moderation can be understood in terms of milliliters of blood loss and the greater IRR for PPH, both of which are clinically meaningful.**
32. Also, is the conclusion that more oxytocin given increases methylation, or rather than increased methylation means more oxytocin is required.
- a. **We detail the different interpretations and future hypotheses in the discussion of the text (lines 515-530). Future work could test if oxytocin exposure causes *OXTR* DNA methylation using a pre-post birth analysis. Alternatively, if *OXTR* DNAm is a relatively stable marker future work could test if it could predict if oxytocin is needed during labor/birth using a prospective method.**
33. Wouldnt the expectation be that increased oxytocin activates receptor expression and therefore loss of DNA methylation?
- a. **In vivo studies with uterine muscle tissue have found that greater oxytocin exposure is associated with lower gene expression over time, as noted in the text. Other studies have demonstrated lower *OXTR* protein availability with greater oxytocin exposure as well (lines 125-127). To our knowledge, *OXTRm* has not been shown to dynamically change in response to gene expression across hours. It is more likely that DNA methylation levels influence the likelihood of gene transcription. Oxytocin exposure probably leads to lower gene transcription/protein availability by factors other than DNA methylation, such as signaling processes leading to removal of transcription factors or placing of transcription repressors on the gene. These are more likely to change in the time scale than DNA methylation.**
34. Overall, conclusions are muddled and not clearly stated, especially in terms of how findings are relevant to clinical applications.
- a. **Clinical applications were listed in the last two paragraphs of the introduction (now lines 107-143) and was available in the discussion section titled “Predicting future oxytocin requirements could improve obstetric care practices” (line 496).**
35. Specific Outstanding questions and issues: How does *oxtr* methylation actually impact oxytocin signaling? Physiologically how is this relevant to the need for increased oxytocin administration, etc.
- a. **This is/was detailed in the introduction (now starting on line 90). First, we explain how prior research has linked *OXTRm* to *OXTR* function and then how *OXTR* function in uterine tissue would potentially lead to differences in uterine-related outcomes during labor and birth. The discussion section goes into further detail about these associations (lines starting at 478).**
36. Show higher blood loss but how does this actually relate to things like, risk of future hemorrhage during future pregnancy, or fatality, etc. state implications in understanding maternal morbidity, but do not discuss morbidity or fatality at all (line 245, page 9)
- a. **We state in the introduction the global statistics around PPH fatality—however, fatality from PPH is probably quite influenced by the available clinical tools for**

controlling and treating hemorrhage and lack of timely access to care as well as underlying conditions that make hemorrhage un-survivable (malnutrition, chronic anemia etc). In the United States, PPH accounts for 11% of pregnancy related deaths, largely because interventions to treat emergencies are more readily available. However, rising PPH rates and PPH related morbidity is the greater concern in our setting. Increased incidence of PPH necessitates more urgent or emergent interventions for more people giving birth—to keep them from dying. As this study was limited to only one birth, we cannot comment on future PPH, however, if a person’s blood derived OXTRm is relatively the same across their reproductive years (which we do not yet know), then their risk may be a feature of their inherent OXTRm characteristics—which could influence more than one birth.

37. Discussion of uterine function and contractility, but not of how oxytm and hemorrhaging are related.
 - a. **The relationship between OXTRm and hemorrhage is through the function of OXTR in causing uterine contraction. Logically, then, if OXTR is less available (due to methylation or other variations), uterine contraction is at potentially less efficient, thus leading to uterine atony and PPH. We are not linking OXTRm to hemorrhage due to other sources of bleeding (i.e. tissue damage) or in regards to mechanisms that contribute to thrombotic/anti-thrombotic pathways.**
38. Connection with hemorrhaging not fully convincing.
 - a. **This study was designed to test OXTRm and PPH, as this has not been reported in any other published works, we have only our data to interpret. We feel the association between the clinical data and the OXTRm is meaningful. The regression analyses all demonstrate that the combination of higher OXTRm and oxytocin administration predicts PPH-related outcomes—despite controlling for the influence of BMI, parity, antibiotic use in labor.**
39. Other cohort issues: Race disparity, BMI differences, gestational diabetes rate, infection and antibiotics. Unclear how these factors were controlled for (stated in line 234).
 - a. **As stated in line 234 (now 354, 363, 436)- these factors were controlled for statistically in the multivariate models. Gestational diabetes did not vary by cases/controls and was therefore not included in the model.**
40. Cohort from aim 1 is mainly multiparous, whereas aim 2 is largely nulliparous. How might this discrepancy impact this study?
 - a. **The case-control study was not designed to assess differences in methylation between parous/nulliparous groups. Nulliparous women were over-represented only slightly in the cases versus controls, nonetheless used parity as a covariate in the regression models. Nulliparous individuals did not differ from multiparous in terms of OXTRm levels (p =0.85), Table 2.**
41. General mentions of uterine contractility and how this relates to oxytocin and hemorrhaging could use a bit more development.
 - a. **Thank you for this comment, we reorganized a good deal of the introduction which we feel helps address many of the comments raised by the reviewer.**

Reviewers' comments:

Reviewer #1 (Remarks to the Author):

Erickson et al. provided thoughtful comments and edits to the manuscript from four detailed reviewers. The introduction still reads rather long and not as focused as it could be, and from the tracked changed version it is not clear what was removed or edited to improve on the original comments. Otherwise, they have been responsive to the remainder of the comments and the manuscript is overall stronger.

Reviewer #3 (Remarks to the Author):

I appreciate the arguments made by the authors and sympathise with them not having the samples to perform the correlation analysis with mRNA levels, but I feel that this is an issue of major importance for the correct interpretation of their data. Without the experiments correlating mRNA with methylation status the data could be best summarised as interesting but not compelling. One senses that this could be an important piece of work when completed and I hope that the authors manage to achieve it.

Reviewer #4 and #5 combined report (Remarks to the Author):

The revised manuscript entitled, "Epigenetic state of oxytocin receptor is associated with exogenous oxytocin needs during human parturition and increased risk of postpartum hemorrhage" highlights a connection between oxytocin receptor expression/methylation, and the need for oxytocin administration during birth, and connects the levels of OXTR methylation to an increased risk for postpartum hemorrhage. This article will contribute to the clinical understanding of risk factors for hemorrhage, and could serve to inform ethical biomarkers predictive of both oxytocin requirements as well as hemorrhage risk during birth.

The careful attention taken to addressing several of the points outlined in the first review is appreciated, and the text throughout flows much better as a result of the changes made. The rationale behind the need for this research is much more apparent as the article presently stands. Additional analyses help further clarify and support findings.

Outstanding minor concerns:

- The separation of study strengths seems out of place and inconsistent with typical article structure. However, we acknowledge that the current placement is an improvement over the original location in the methods section.
- We would like to suggest adding a phrase to the limitation section stating that parity (nulliparous, parous, multiparous) is a potential caveat in this data set. We raised this suggestion (regarding multiparity and age), but this was not fully resolved after revision and rebuttal. It has been established that both age and parity influence women's health, and thus mentioning could help guide future studies when considering how to collect data and whom to enroll.

- We would also suggest, in the limitations, to include a note about diversity of patients, particularly given that race is associated with poor quality of care and increased maternal death during pregnancy.

Editor's Comments:

The major outstanding issue is the main claim of the manuscript and how this is framed. Clearly your results are not supportive of OXTR levels of expression being a biomarker for hemorrhage. However, you are able to suggest that changes in DNA methylation levels around/at the OXTR gene locus can be used as biomarkers of hemorrhage, independently of the levels of mRNA. In addition to framing the manuscript in this way, you should also highlight that the next steps are to look at mRNA and also to devise a strategy to enable evaluation of DNA methylation alterations to define patients at risk of hemorrhage. Providing a proposed path for how such information will actually be used as a biomarker would be helpful.

We therefore invite you to revise and resubmit your manuscript, taking into account the above point and the remaining issues raised by the reviewers. Please highlight all changes in the manuscript text file.

Thank you for these comments: in response to these requests we have done the following:

- Revised the introduction/ trimmed paragraphs
- Added a phrase in the abstract about need to link findings to gene expression in future work (line 34)
- Added a line in the methods (299) about not preserving tissue in RNA safe medium.
- Line 606: added another line on future directions for OXTR expression studies and functional changes in uterine tissue
- Line 608: Further reinforces that we only looked at OXTRm
- Line 634: a comment to sum up that our study was indicating that OXTRm was relevant as a biomarker and requires further functional studies of tissue level differences
- Line 663: moved/updated a sentence about prospective testing of OXTRm to use this as a biomarker in clinical translational studies
- Lines 691-694: added lines for recommended future studies to consider maternal age, parity and racial/ethnic diversity of the sample.
- Line 716: we qualify our findings as 'interesting' in the conclusion

Reviewer #1 (Remarks to the Author):

Erickson et al. provided thoughtful comments and edits to the manuscript from four detailed reviewers. The introduction still reads rather long and not as focused as it could be, and from the tracked changed version it is not clear what was removed or edited to improve on the original comments. Otherwise, they have been response to

the remainder of the comments and the manuscript is overall stronger.

Thank you for the comments, we have further edited the introduction, cutting about 1/3 page of text and a bit of reorganization of the paragraphs to help with the focus.

Reviewer #3 (Remarks to the Author):

I appreciate the arguments made by the authors and sympathise with them not having the samples to perform the correlation analysis with mRNA levels, but I feel that this is an issue of major importance for the correct interpretation of their data. Without the experiments correlating mRNA with methylation status the data could be best summarised as interesting but not compelling. One senses that this could be an important piece of work when completed and I hope that the authors manage to achieve it.

Thank you, we also feel there is more work to be done and look forward to taking those next steps.

Reviewer #4 and #5 combined report (Remarks to the Author):

The revised manuscript entitled, “Epigenetic state of oxytocin receptor is associated with exogenous oxytocin needs during human parturition and increased risk of postpartum hemorrhage” highlights a connection between oxytocin receptor expression/methylation, and the need for oxytocin administration during birth, and connects the levels of OXTR methylation to an increased risk for postpartum hemorrhage. This article will contribute to the clinical understanding of risk factors for hemorrhage, and could serve to inform ethical biomarkers predictive of both oxytocin requirements as well as hemorrhage risk during birth.

The careful attention taking to addressing several of the points outlined in the first review is appreciated, and the text throughout flows much better as a result of the changes made. The rationale behind the need for this research is much more apparent as the article presently stands. Additional analyses help further clarify and support findings.

Thank you

Outstanding minor concerns:

- The separation of study strengths seems out of place and inconsistent with typical article structure. However, we acknowledge that the current placement is an improvement over the original location in the methods section.

Thank you, we have limitations and strengths listed in the last part of the discussion, we are open to editors' comments if we should change the location.

- We would like to suggest adding a phrase to the limitation section stating that parity (nulliparous, parous, multiparous) is a potential caveat in this data set. We raised this suggestion (regarding multiparity and age), but this was not fully resolved after revision and rebuttal. It has been established that both age and parity influence women's health, and thus mentioning could help guide future studies when considering how to collect data and whom to enroll.
- We would also suggest, in the limitations, to include a note about diversity of patients, particularly given that race is associated with poor quality of care and increased maternal death during pregnancy.

We have added text into the limitations section to address the above two bullet points for recommendations for future research

REVIEWERS' COMMENTS:

Reviewer #4 and #5 (Remarks to the Author):

The authors have addressed all of the major points.